# Illustrated Landmark Graphs for Long-Horizon Policy Learning

**Christopher Watson**                                     *ccwatson@seas.upenn.edu*
*Department of Computer and Information Science*
*University of Pennsylvania*

**Arjun Krishna**                                             *arjk@seas.upenn.edu*
*Department of Computer and Information Science*
*University of Pennsylvania*

**Rajeev Alur**                                               *alur@cis.upenn.edu*
*Department of Computer and Information Science*
*University of Pennsylvania*

**Dinesh Jayaraman**                                       *dineshj@seas.upenn.edu*
*Department of Computer and Information Science*
*University of Pennsylvania*

**Reviewed on OpenReview:** *https://openreview.net/forum?id=0AOUWC4ss8*

## Abstract

Applying learning-based approaches to long-horizon sequential decision-making tasks requires a human teacher to carefully craft reward functions or curate demonstrations to elicit desired behaviors. To simplify this, we first introduce an alternative form of task-specification, Illustrated Landmark Graph (ILG), that represents the task as a directed graph where each vertex corresponds to a region of the state space (a *landmark*), and each edge represents an easier to achieve sub-task. A landmark in the ILG is conveyed to the agent through a few illustrative examples grounded in the agent's observation space. Second, we propose ILG-Learn, a human in the loop algorithm that interleaves planning over the ILG and sub-task policy learning. ILG-Learn adaptively plans through the ILG by relying on the human teacher's feedback to estimate the success rates of learned policies. We conduct experiments on long-horizon block stacking and point maze navigation tasks, and find that our approach achieves considerably higher success rates ($\approx 50\%$ improvement) compared to hierarchical reinforcement learning and imitation learning baselines. Additionally, we highlight how the flexibility of the ILG specification allows the agent to learn a sequence of sub-tasks that is better suited to its limited capabilities.

## 1 Introduction

How can a human best teach an agent to perform a new long horizon task? The two most popular classes of approaches today are reinforcement learning (RL) and imitation learning. In RL, (Sutton & Barto, 2018) the teacher specifies the task via a reward function that assigns higher scores to more desirable environment configurations. The agent then interacts with the environment to learn how to achieve high rewards. For long-horizon tasks, the efficiency of this trial-and-error exploration depends intimately on the human teacher's ability to design a *well-shaped* reward function that encourages incremental progress towards the final goal (Laud, 2004; Ng et al., 1999; Sowerby et al., 2022; Gupta et al., 2022). As a concrete example, let us consider the block stacking task (`StackChoice`) shown in Figure 1. The task is to build a tower; the agent may choose to place block A (red) on block B (green) or *vice versa*. The task itself is fully specified by a

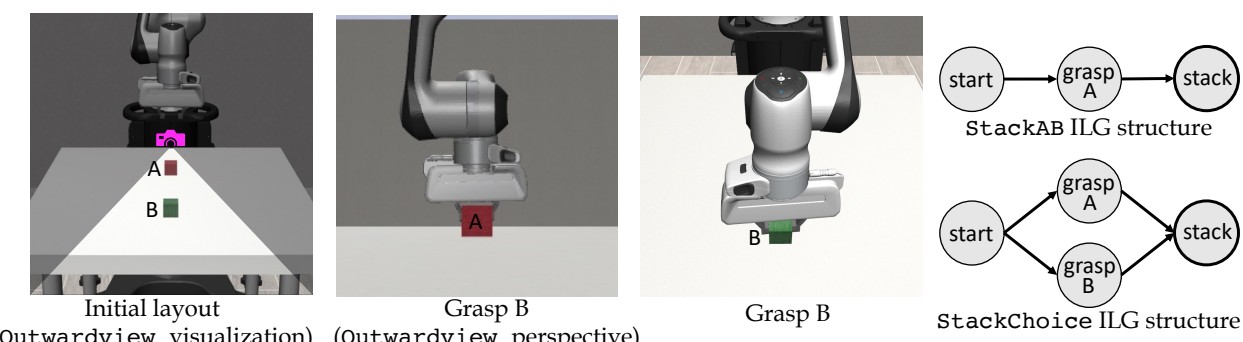

Figure 1: **Left:** Illustrates the `Stack` task. At the start of each episode, block A (red) starts closer to the robot than block B (green). Both blocks initialize at randomized positions along the centerline of the table. In the `Outwardview` condition, block A initially occludes block B. **Right:** ILG structure for variations of the `Stack` task. The `StackBA` ILG is defined analogously.

simple "sparse" reward function that is 1 when the blocks are stacked and 0 otherwise. However, in practice, the human teacher must write a more sophisticated reward function that provides "dense" rewards during task execution. For example, it might incorporate small positive rewards for reaching various milestones towards the task: moving the gripper close to the block, grasping, lifting, aligning, and finally placing the block. To write such a reward function, the human teacher must be able to interpret the sensor readings available to the learner and prudently balance the weight given to each term in the reward function. In general, reward engineering is difficult, error-prone and requires considerable expertise, (Booth et al., 2023; Amodei et al., 2016; Skalse et al., 2022) limiting RL's applicability to long-horizon tasks.

Imitation learning (Hussein et al., 2017; Zare et al., 2023; Ravichandar et al., 2020) allows the human to teach the agent by demonstrating desired behavior. Compared to RL, imitation learning shifts the teaching burden from reward engineering to demonstration. Although demonstrations are an intuitive form of task specification, providing high-quality demonstrations may not always be feasible (Ravichandar et al., 2020). The teacher might not know how best to perform a task, or how to manually operate the agent to do so, e.g. there may be no good interfaces to manually operate a biped robot to run smoothly. Even without these problems, demonstrations can be cumbersome to provide: in the `StackChoice` task, the teacher would need to teleoperate the robot and gather many demonstrations to adequately cover the states that the agent may encounter during deployment. Finally, demonstrations must account for the learner's limitations: e.g., stacking the blocks in one order may be easier for the robot to perform than the other, because of limitations on its sensing, perception, or actuation abilities – which are sometimes hard to characterize in advance.

Reward functions and demonstrations occupy opposite ends of a design spectrum: a reward function specifies *what* must be achieved (perhaps without much intermediate guidance), while a demonstration specifies *how* to achieve it (perhaps too prescriptively). Ideally, the teacher could provide a specification that balances the declarative nature of a reward function (which exposes opportunities for the learner to customize its approach according to its capabilities) with the prescriptive nature of a demonstration (which provides fine-grained guidance). To address this gap, we propose the *illustrated landmark graph (ILG)* as a form of long-horizon task specification that is inspired by the use of temporal logic to structure reinforcement learning (Li et al., 2017; Jothimurugan et al., 2021). Each vertex of an ILG represents an intermediate *landmark*, which is a subset of the environment's state space, analogous to an *atomic proposition* of a temporal logic specification. To show the meaning of a landmark to the learner, the teacher must be able to provide a handful of *illustrative observations* drawn from states within the landmark. Each directed edge $(u, v)$ in the ILG represents the *edge task* of transitioning the environment from the landmark represented by $u$ to the landmark represented by $v$. Each sink vertex represents a *final landmark*. The long horizon task is to reach any final landmark, passing through intermediate landmarks on the way. Importantly, the ILG can contain multiple paths to a final landmark, exposing opportunities for the learner find a plan that suits its capabilities. Returning to our block stacking task, a human teacher can identify that grasping a block before placing it is a useful landmark and can communicate this to the learner via the `StackChoice` ILG shown in Figure 1.

To leverage the ILG for policy learning, we propose ILG-Learn. At a high level, ILG-Learn interleaves Dijkstra-style planning over the ILG with *example-based control*, (Fu et al., 2018; Singh et al., 2019; Eysenbach et al., 2021) a lightweight alternative to imitation learning that uses goal observations instead of full-length demonstrations, to learn an *edge policy* for each edge task. This allows the learner to benefit from temporal abstraction, as in hierarchical reinforcement learning, (Hutsebaut-Buysse et al., 2022) without the need for reward engineering. Concretely, to train an edge policy for "*start → grasp B*", the teacher would provide illustrative observations by manipulating the gripper to grasp block B, then letting the learner observe the grasp with its onboard sensors. Example-based control lets the learner use such illustrative observations in lieu of a reward function as it interacts with the environment to learn a control policy to grasp block B. Beyond providing the ILG and its associated illustrative observations, the ILG-Learn teacher has to only respond to binary success/failure queries during training to inform the learner's graph search. This intuitive teacher-learner interface allows the learner to benefit from exploration without requiring reward engineering.

A key benefit of the ILG specification is the ability to represent multiple paths to overall success. The `StackChoice` ILG includes multiple paths to the final landmark ("*stack*")—depending on the learner's capabilities it may be easier stack the blocks in one order or the other. For example, if the agent were to perceive the world using an outward-facing camera mounted opposite the robot on the surface of the table (as shown in Figure 1), block B would initially be occluded, making it easier to start by grasping the (unoccluded) block A. By querying the teacher for success/failure feedback, the ILG-Learn learner can focus exploration along the most promising paths. Our experiments show that ILG-Learn adapts to the occlusion condition and learns a policy that grasps the unoccluded block first. This capacity for train-time adaptation reduces the burden on the human teacher to predict which approach will be most feasible for the learner. In this way, the ILG differs from the symbolic abstractions used for Task & Motion Planning, (Zhao et al., 2024) in which the human must define which transitions are feasible before planning begins.

Another benefit of the ILG is that the teacher can choose a *landmark density* (the number of intermediate vertices along a path in the ILG) that is dense enough to guide efficient exploration yet not so dense as to be overly prescriptive. At one extreme, the teacher could provide a landmark graph that comprises a single edge, thus recovering the free-form exploration of ILG-Learn's underlying example-based control algorithm. At the other extreme, the teacher could identify many intermediate landmarks to tightly scaffold exploration. The ideal landmark density depends on the task of interest; we believe that the intuitive structure of the ILG will enable the teacher to apply domain knowledge to select an appropriate landmark density without concerning themselves with the low-level details of the robot's perception and motor capabilities. Our experiments show that ILG-Learn outperforms baselines that use end-to-end example-based control (which embodies minimal landmark density) and behavior cloning (which uses full-length demonstrations). In summary:

- We introduce the ILG as a form of task specification that allows a human teacher to decompose a task into a series of easier to achieve landmarks, thereby providing intermediate guidance without reward engineering, full length demonstration, or explicit knowledge of the learner's capabilities.

- We propose ILG-Learn, a learning algorithm that leverages the ILG's interpretable-by-design nature to allow the teacher to provide lightweight guidance as the learner learns policies and a plan to suit its unique capabilities.

- We empirically show that providing a suitable ILG enables ILG-Learn to learn long-horizon policies that achieve significantly ($\sim 50\%$) higher success rates compared to approaches that do not receive structured guidance (RCE and behavior cloning). Additionally, we demonstrate how a multi-path ILG facilitates learning when the teacher cannot fully anticipate the learner's capabilities.

## 2  Related work

**Hierarchical reinforcement learning, Task & Motion Planning.** Hierarchical reinforcement learning (HRL) (Hutsebaut-Buysse et al., 2022; Pateria et al., 2021) incorporates temporal abstraction into reinforcement learning, effectively decomposing a long horizon task into easier to achieve sub-tasks. Prior works (Kulkarni et al., 2016; Nachum et al., 2018) have shown that incorporating some human priors for

temporal task decomposition enables RL algorithms to find useful abstractions to solve long-horizon tasks. The ILG specification format allows the human teacher to provide such abstractions; at a high level, each edge in the ILG is analogous to an *option* in the options framework (Sutton et al., 1999). Unlike existing HRL techniques, ILG-Learn uses example-based control and teacher-learner interaction to avoid the need for reward engineering and explicit specification in terms of environment states.

In robotics, Task & Motion Planning (Garrett et al., 2021; Zhao et al., 2024) is a popular framework to decompose long-horizon tasks into high-level symbolic task planning and low-level continuous motion optimization. Typically, a TAMP system designer must specify task-relevant abstractions such as motion primitives, their affordances, and constraints, etc., to synthesize task and motion plans under the assumption that the required state is fully observable and the effects of chosen sub-task and action sequences can be modeled deterministically. Contrary to this, to create an ILG the teacher only needs to specify various potentially feasible sequences of sub-tasks and ground them via illustrative observations. During training, ILG-Learn uses the ILG to scaffold exploration and discovers a sequence of landmarks that the agent can feasibly achieve based on its capabilities. This directed exploration to arrive at the effective task plan is reminiscent of Go-Explore (Ecoffet et al., 2021). ILG-Learn differs from Go-Explore in that landmarks serve as the task specification (avoiding the need for reward design), include human-defined relationships (in the form of the ILG), and are illustrated to the learner via teacher-provided observations (rather than reached organically).

**Specification-guided reinforcement learning and reward machines.** ILG-Learn's use of the human-specified ILG is inspired by *specification-guided reinforcement learning*, in which a human provides a temporal logic specification that can be used to structure hierarchical reinforcement learning (Jothimurugan et al., 2021; Araki et al., 2021) or reward function generation (Li et al., 2017; Bozkurt et al., 2020). The main benefit of specification-guided reinforcement learning is that some (or all) reward engineering effort can be replaced with logical specification, which may be more intuitive and less error prone. A specification-derived reward function can be expressed as a *reward machine* (Toro Icarte et al., 2022), which is a finite-state automaton whose current state determines the reward function. Both the ILG and the reward machine expose the symbolic structure of a multi-step task to a learner. Unlike an ILG, however, a reward machine does not include a distinguished "final goal" state; maximizing accumulated reward may or may not require reaching a particular reward machine state. The ILG exposes the high-level objective (reach a final landmark) to the learner, who must combine graph-based planning and example-based control to obtain local guidance. On the other hand, a reward machine immediately exposes local guidance to the learner (in the form of the initial reward machine state's reward function) and the learner may leverage the reward machine's symbolic structure to guide exploration as it seeks to maximize accumulated reward.

ILG-Learn's use of the ILG is directly inspired by DiRL's (Jothimurugan et al., 2021) use of the analogous *abstract graph*, which is built according to the structure of a temporal logic specification. ILG-Learn and DiRL both use the specification's structure to scaffold train-time planning; this contrasts with the approach of Yalcinkaya et al. (2024), which uses a neural network to produce an embedding of the specification serves as input to a goal-conditioned policy. The ILG-definable specifications are subsumed by linear temporal logic; ILG-Learn can equivalently be seen as a specification-guided reinforcement learning algorithm in which the learner must infer the semantics of each atomic predicate (landmark) from a set of illustrative observations and active success/failure queries.

**Imitation learning and example-based control.** As we argued in Section 1, the price of imitation learning's intuitive teacher-learner interface is rigidity: the learner is vulnerable to compounding deviations from the teacher's demonstrations (Ross et al., 2011) and may not enjoy the flexibility to adjust its approach to suit its own unique capabilities. To address these challenges, *inverse reinforcement learning* (Arora & Doshi, 2021; Adams et al., 2022) gleans the demonstrator's intent, allowing the agent to learn a policy through environmental interaction rather than rote memorization. In the context of long horizon policy learning, UVD (Zhang et al., 2024) and Relay Imitation Learning (Gupta et al., 2020) identify subgoals within demonstrations that can be used for compositional imitation inspired by goal-conditioned RL. Compared to ILG-Learn, these approaches remove the need for the teacher to identify landmarks, but in so doing reduce

the teacher's ability to scaffold exploration and require the teacher to provide demonstrations rather than illustrative observations.

*Example-based control* (Fu et al., 2018; Singh et al., 2019; Eysenbach et al., 2021) allows the teacher to provide single-timestep examples of the environment state after task completion rather than full-length demonstrations. It is often significantly easier for a human to provide a single-timestep example than a full demonstration, and sparse guidance allows the learner more flexibility. However, for long-horizon tasks, such low landmark density may not provide the learner enough guidance to complete the task. At a high level, our proposed method, ILG-Learn, scales example-based control to long-horizon tasks by incorporating human exploration biases in the form of intermediate landmarks. We use the example-based control algorithm RCE (Eysenbach et al., 2021) to learn each edge policy that transitions the environment between landmarks.

**Human-in-the-loop policy learning.** Reinforcement learning with human feedback (RLHF) (Kaufmann et al., 2024; Arzate Cruz & Igarashi, 2020; Najar & Chetouani, 2021) elicits guidance from a human during training to reduce (or completely eliminate) reliance upon the environment's reward function. Some recent applications of RLHF to robotics query human preference over states to discover subgoals for HRL (Zhou et al., 2024) or goal-conditioned exploration (Villasevil et al., 2023). Instead of using human preference to discover landmarks, ILG-Learn requires the teacher to choose which landmarks to include in the ILG before learning begins. ILG-Learn then incorporates human feedback in the form of success/failure queries to direct exploration towards paths that cater to the learner's capabilities. ILG-Learn's use of active queries to steer the learner away from undesirable behavior is inspired by VICE-RAQ, (Singh et al., 2019) an example-based control algorithm that targets success/failure queries on states that closely resemble human-provided examples, which prevents an inferred reward function from encouraging progress towards spurious goals.

Human feedback can also be used to facilitate imitation learning. For example, HI-IRL (Pan & Shen, 2018) requires the human to provide subgoal states alongside a full-length demonstration. During training, the learner can ask for focused demonstrations of particularly hard-to-learn subgoals. Another recent approach, Thrifty DAgger (Hoque et al., 2021) quantifies *novelty* and *risk* to make the most of a limited human interaction budget. Another approach in the DAgger (Ross et al., 2011) family, RLIF (Luo et al., 2024) lets the human intervene during training to tell the learner when it behaves unsatisfactorily and provide a short demonstration of better behavior. The aforementioned techniques begin with a full-length demonstration of the task, which contrasts with our ILG's ability to express multiple paths to success. In a similar vein, Memarian et al. (2020) combine automata learning and inverse reinforcement learning to discover a subgoal decomposition alongside a policy to complete a long-horizon task. Their approach allows the learner to ask for specific guidance while discovering the subgoal decomposition but is not immediately applicable to continuous-state environments with difficult control tasks.

## 3 Illustrated landmark graphs

We introduce the *illustrated landmark graph (ILG)* as a form of long-horizon task specification. A key feature of the ILG is that the teacher communicates the meaning of each landmark to the learner using *illustrative observations* and does not need to provide explicit definitions in terms of the environment's state. In Section 3.1 we define the environment and policy model before defining the ILG specification format in Section 3.2. The ILG is a useful form of specification because of the way it facilitates teacher-learner interaction; we describe our proposed interaction interface in Section 3.3.

### 3.1 Preliminaries

As is standard in RL, we assume the agent interacts with an environment that can be expressed as a Markov Decision Process (MDP). An MDP comprises a continuous set of states $\mathcal{S}$, a continuous set of actions $\mathcal{A}$, dynamics governed by the transition probability distribution $p(\boldsymbol{s}_{t+1} \mid \boldsymbol{s}_t, \boldsymbol{a}_t)$, and a starting state distribution $\eta$. Instead of specifying the task via a reward function (as is commonly done in MDPs), we will use an ILG as the task specification and incorporate human interaction to guide the learner.

At each timestep $t$, the agent receives an observation $o(\boldsymbol{s}_t) \in \Omega$ where $\Omega$ is the observation space. The observation function $o : \mathcal{S} \to \Omega$ captures the learner's perception capabilities. In a real-world setting, $o(\boldsymbol{s}_t)$ may represent raw sensor readings or the outputs of a state-estimation module. A policy is a function $\pi : (\Omega \times \mathcal{A})^* \times \Omega \to \Delta(\mathcal{A})$ that maps a trace of observations and actions to a distribution over next actions.

### 3.2 ILG and satisfaction

Each vertex of an ILG represents a landmark, which is a subset of the state space $\mathcal{S}$, and each edge in the the ILG represents the *edge task* of transitioning between two landmarks. The ILG must have at least one sink vertex; each sink corresponds to a *final landmark*. The agent's goal is to reach a final landmark.

Formally, an ILG is a tuple $(U, E, u_0, \beta)$ with vertex set $U$, directed edge set $E$, distinguished source vertex $u_0$, and landmark map $\beta : U \to 2^{\mathcal{S}}$ that maps each vertex to the set of states that comprise the corresponding landmark. An ILG must have at least one sink vertex. A trajectory $\xi = \boldsymbol{s}_0 \boldsymbol{a}_0 \boldsymbol{s}_1 \boldsymbol{a}_1 \dots$ satisfies the landmark graph $(U, E, u_0, \beta)$ iff there exists a sink vertex $u \in U$ and a time $t$ such that $\boldsymbol{s}_t \in \beta(u)$.

Figure 1 shows the structure of the `StackChoice` ILG. There are two source-to-sink paths: "$start \to grasp\ A \to stack$" and "$start \to grasp\ B \to stack$." Any trajectory that passes through a state in the landmark represented by the sink vertex "$stack$" satisfies the specification. The vertices "$grasp\ A$" and "$grasp\ B$" provide intermediate guidance to a policy learner. This guidance takes the form of teacher-learner interaction during training as detailed in the following subsection.

### 3.3 Interaction and illustration

As is common in RL, we assume that the learner does not begin with knowledge of the environment's transition probabilities; it must learn about environment dynamics through interaction. Similarly, we assume that the learner does not have explicit access to the ILG specification's landmark map $\beta : U \to 2^{\mathcal{S}}$. During training, the learner begins with access to the *structure* $(U, E, u_0)$ of the ILG $(U, E, u_0, \beta)$. In order to obtain guidance toward the landmark represented by a vertex $u$, the learner must interact with the teacher to request *illustrative observations* drawn from states within $\beta(u)$. The learner may also request success/failure feedback from the teacher to see if the current state of the environment lies within a particular landmark.

During training, the learner interacts with the teacher and with the environment using the following procedures. Training proceeds as a sequence of episodes. At each timestep $t$ within an episode the learner receives an observation $o(\boldsymbol{s}_t)$ and may:

- `reset()` Reset the environment to a state $\boldsymbol{s}_0 \sim \eta$. The learner receives the observation $o(\boldsymbol{s}_0)$.

- `step(`$\boldsymbol{a}_t$`)` Provide an action $\boldsymbol{a}_t$ to the environment, which causes the environment to transition to a state $\boldsymbol{s}_{t+1} \sim p(\cdot|\boldsymbol{s}_t, \boldsymbol{a}_t)$. The learner receives the observation $o(\boldsymbol{s}_{t+1})$.

- `requestIllustration(`$u, \rho$`)` where $u \in U$ is the vertex of the ILG to be illustrated and the second argument $\rho \in U^*$ lets the the learner tell the teacher which path it wishes to extend to $u$. In response, the teacher provides a dataset of illustrative observations $\mathcal{O}_u \subset \{o(\boldsymbol{s}) \in \Omega \,|\, \boldsymbol{s} \in \beta(u)\}$ of illustrative observations of states within $\beta(u)$. Although the path $\rho$ does not affect the contract that the teacher must uphold, it may help the teacher choose a useful set of illustrative observations: in the `StackChoice` example, a good teacher would respond to `requestIllustration(`$stack, start \to grasp\ A$`)` with examples of block A stacked on top of block B (not of B stacked on A).

- `querySuccess(`$u$`)` where $u \in U$ is a vertex of the ILG. The teacher provides binary success/failure feedback of whether the current environment state $\boldsymbol{s}_t$ is in the landmark $\beta(u)$.

Given an MDP $\mathcal{M}$, observation function $o$, and ILG $\mathcal{G}$, the learner tries to learn a policy $\pi$ that maximizes the probability that a trajectory $\xi$ drawn from $\pi$ interacting with $\mathcal{M}$ while receiving observations according to $o$ satisfies $\mathcal{G}$.

Importantly, the teacher is never asked to explicitly define the landmark map $\beta$. Instead, the teacher must be able to provide a set of illustrative observations for each landmark and serve as an oracle for `querySuccess`.

The teacher can provide a set of illustrative observations by, for example, physically positioning a robot and allowing its sensors to perceive the environment. This intuitive interaction interface allows the teacher to guide the learner without needing to understand the low-level details of the learner's perception capabilities.

Underlying the ILG's usefulness is its customizability. To permit multiple approaches to complete the task, the human can include multiple paths that lead to a final landmark. In addition to using branching to expose options for high-level planning, the teacher can adjust the *landmark density* (the number of intermediate landmarks) along each path to provide the appropriate granularity of guidance. In the context of policy learning, low landmark density cedes resolution of low-level control details to the underlying learning algorithm. This is both convenient for the teacher and can result in well-optimized motion if the learner can successfully transition between landmarks. On the other hand, excessively low landmark density can render policy learning intractable, as the underlying learning algorithm is left with insufficient exploration bias.

## 4    Learning algorithm

We propose ILG-Learn, a human-in-the-loop approach to policy learning for ILG specifications. At a high level, ILG-Learn performs the following steps:

- **Policy learning.** For an edge $(u, v)$ of the ILG, the learner uses example-based control to learn an edge policy $\pi_{(u,v)}$ that tries to transition the environment from the landmark represented by $u$ to the landmark represented by $v$.

- **Success estimation.** Given a learned path policy (a sequence of edge policies), the learner queries the teacher for binary success/failure feedback to estimate the success probability of the policy.

- **Planning.** The learner tries to find a source-to-sink path of minimal cost, where cost is defined as the negative logarithm of the associated path policy's success probability.

At the outset, the learner receives the structure $(U, E, u_0)$ of the ILG specification $(U, E, u_0, \beta)$ but does not know how to reach any landmark. By interacting with the environment and the teacher (through the interface described in Section 3.3) the learner learns policies to reach intermediate landmarks and a plan to satisfy the ILG specification by reaching a final landmark (represented by a sink vertex). ILG-Learn iteratively builds a tree of lowest-cost known paths from the source to the other vertices of the ILG. Since the cost of a path depends on the success probability of the learned edge policies, ILG-Learn interleaves graph search and policy learning to obtain the path costs on the fly during training. Learning proceeds as a series of *learning intervals*, each of which focuses on a single edge of the ILG. Once ILG-Learn finds the lowest cost path $\rho$ to a sink vertex, ILG-Learn returns the plan $\rho$ and the associated path policy.

Algorithm 1 sketches the full ILG-Learn algorithm and Figure 2 illustrates the steps of a learning interval. In the following textual description we will denote the ILG specification as $(U, E, u_0, \beta)$ and assume the learner has access to the environment $\mathcal{M}$ and the human teacher through the interface described in Section 3.3. To make our description concrete, we will use the `StackChoice` task introduced in Section 1 as a running example. The structure $(U, E, u_0)$ of the `StackChoice` ILG is shown in Figure 1.

---

ILG-Learn hyperparameters

**illustrationCount**: illustrative observations per request.
**episodeLength**: fixed horizon for edge policy training.
**intervalLength**: environment steps per learning interval (exclusive of those used during resets).
**estimationQueries**: # rollouts (each with a success/failure query) at end of each learning interval.
**successThreshold**: lower bound on success probability for an edge to be considered learned.
**intervalsLimit**: max # of learning intervals per edge.

---

**Exploration order (`selectEdge`).** ILG-Learn's exploration order is determined by the `selectEdge` subroutine. At the outset, ILG-Learn's tree of known paths contains only the "start" vertex $u_0$. To extend this

---

**Algorithm 1:** ILG-Learn

---

**Input:** ILG structure $(U, E, u_0)$, access to MDP $\mathcal{M}$ via `reset` and `step`, access to human teacher via `requestIllustration` and `querySuccess`.

**Output:** Path $\rho$ and associated path policy

$\rho_{u_0}$, $\pi_{u_0}$, $reachProb_{u_0} \leftarrow [\,]$, None, 1;
$reachProb_u \leftarrow 0 \quad \forall u \in V \setminus \{u_0\}$;
**while** *selectEdge() returns* $(u, v)$ **do**
    $\mathcal{O}(v) \leftarrow$ `requestIllustration`$(v, \rho_u)$;
    $\pi_{(u,v)} \leftarrow$ `learnPolicy`$(\mathcal{O}_v, \rho_u, \pi_u)$;
    $\pi \leftarrow$ `sequencePolicies`$(\pi_u, \pi_{(u,v)})$;
    $successProb \leftarrow$ `estimateProbability`$(v, \pi)$;
    **if** $successProb > reachProb_v$ **then**
        $reachProb_v \leftarrow successProb$;
        $\rho_v \leftarrow \rho_u \circ (u, v)$;
        $\pi_v \leftarrow \pi$;
$u \leftarrow \text{argmax}_{u \in sinkVertices(\mathcal{G})}(reachProb_u)$;
**return** $\rho_u$, $\pi_u$

---

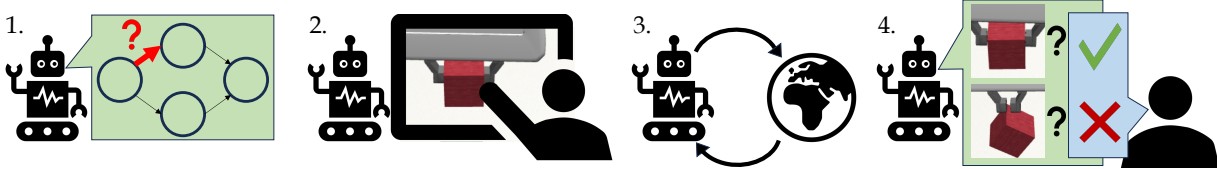

Figure 2: Stages of a learning interval: (1) The learner selects an edge and requests illustrative observations to guide learning. (2) The human teacher provides illustrative observations of the destination landmark. (3) The learner uses example-based control to learn an *edge policy* from environmental interaction. (4) The learner executes the edge policy and queries the teacher for success/failure feedback.

tree, `selectEdge` chooses an edge leaving the tree for which to learn a control policy. ILG-Learn implements a fine-grained interleaving (mediated by the **successThreshold** and **intervalsLimit** hyperparameters) of policy and plan learning to accelerate learning. For details, see Appendix A.

Let us suppose that `selectEdge` chooses the edge $(u, v) = (start, grasp\ A)$. This marks the start of a *learning interval* dedicated to learning the edge policy $\pi_{(u,v)}$.

**Human guidance (`requestGoals`).** After selecting an edge $(u, v)$, ILG-Learn asks the human teacher to provide a dataset $\mathcal{O}_v$ containing **illustrationCount** illustrative observations drawn from states within $\beta(v)$. In our example, the teacher would provide a set of observations that illustrate "*grasp A*".

**Policy learning (`learnPolicy`).** ILG-Learn runs example based control for **intervalLength** environment steps (structured as episodes of **episodeLength** timesteps) to learn an edge policy $\pi_{(u,v)}$. This learning process does not require interaction with the human teacher.

In our example, the learner would interact with the environment to learn a policy that tries to reach states that yield observations that look similar to the illustrative observations of the "*grasp A*" landmark. This similarity is quantified by the example-based control algorithm. Our implementation uses RCE, (Eysenbach et al., 2021) which is inspired by actor-critic reinforcement learning. In RCE, the critic network is trained to predict a time-discounted success probability for observation-action pairs. Success is defined as a regression task: the success probability of illustrative observations is regressed to 1 while other observations (collected in a replay buffer) are assumed to not represent success. The critic's scores are used to optimize the policy, similarly to the soft actor critic RL algorithm (Haarnoja et al., 2017). Since this is the first edge along a path, each learning episode starts from the environment's reset distribution.

In a future learning interval, ILG-Learn may try to learn the next step of the path, namely, how to stack block A on top of block B (assuming block A is already grasped). Since policy learning is sensitive to the starting state distribution, ILG-Learn needs to start learning $\pi_{(u,v)}$ from the distribution induced by the (already learned) path policy $\pi_u$. So in the case of learning to stack block A on block B, each episode would start by executing the ($start, grasp\ A$) policy (for the same fixed horizon **episodeLength**) to reach the appropriate starting distribution for the ($grasp\ A, stack$) edge task.[1]

**Success Estimation (`estimateProbability`)**  The last step of each learning interval is to estimate the probability that the learned edge policy $\pi_{(u,v)}$ successfully reaches $\beta(v)$. Just as it was important to start training $\pi_{(u,v)}$ from the state distribution reached by executing $\pi_u$, it is important to evaluate $\pi_{(u,v)}$ from the distribution reached by $\pi_u$. The `estimateProbability` subroutine executes **estimationEpisodes** additional rollouts of the sequenced policy, using `querySuccess(`$u$`)` at the last timestep of each rollout, to ask the human teacher whether the rollout was successful. If this empirical path probability exceeds that of the best known path $\rho_v$ to $v$, ILG-Learn updates the best known path and associated path policy $\pi_v$. If the empirical probability exceeds **successThreshold** (or the per-edge max number of learning intervals **intervalsLimit** is reached) then ILG-Learn will consider the edge fully explored. At this point, the path cost (the negative logarithm of the success probability) is solidified. No more learning intervals can be allocated to $(u, v)$. Future learning intervals can now be allocated to edges that leave $v$.

Since success estimation only occurs at the end of each learning interval, the **intervalLength** hyperparameter mediates a tradeoff between learning efficiency in terms of total environment steps and learning efficiency in terms of human annotation burden. A short **intervalLength** means that the `estimateProbability` subroutine is executed more frequently, which reduces the likelihood that superfluous training effort will be invested in an edge policy. However, `estimateProbability` requires the teacher to respond to success/failure queries. We provide advice for selecting ILG-Learn's hyperparameters in Appendix B.

**Termination.**  Training is complete when the `selectEdge` subroutine returns *None*, indicating that ILG-Learn has found the lowest-cost path in the ILG and the associated path policy. Otherwise, a new learning interval will commence, focusing exploration along the edge chosen by `selectEdge`.

## 5  Experiments

Our experiments show that ILG-Learn can learn policies to successfully complete long horizon tasks. To better understand the importance of allowing the teacher to choose the ILG's landmark density we evaluate ILG-Learn against RCE, a state of the art example-based control algorithm that does not use intermediate landmarks, and behavior cloning (BC), an imitation learning algorithm that receives full-length demonstrations. We include two variants of BC, one which predicts actions using an MLP that is architecturally similar to ILG-Learn and RCE, as well as the modern *diffusion policy* (Chi et al., 2023) architecture. Since the ILG task specification is new, there do not exist direct analogs of ILG-Learn to baseline against; rather, RCE's examples and BC's demonstrations represent two alternatives to ILG specifications.

We also compare against *hierarchical reinforcement learning with reward machines (HRM)*, (Toro Icarte et al., 2022) an HRL algorithm that exploits the structure of a handcrafted *reward machine* that provides dense rewards. Our reward machine definitions mirror the structure of the ILG for each task; our experiments explore whether ILG-Learn's use of example-based control to learn each edge task can rival or exceed the efficacy of reinforcement learning with dense rewards.

Finally, we investigate whether ILG-Learn can discover the path through multi-path (branching) ILG that suits the learner's capabilities. Such train-time adaptation would reduce the burden on the teacher to predict the best approach to task completion *a priori*. We conduct our experiments in simulation using the following environments:

---

[1]Steps used during resets do not contribute to the **intervalLength** steps per learning interval, however in our experimental evaluation we limit the *total* number of environment steps to 10 million.

**Stack.** Our `Stack` family of environments is a customized `robosuite` (Zhu et al., 2020) environment that simulates a 7-DoF Franka Panda arm that receives 55-dimensional state observations and uses a 7-dimensional action space (which represents an operational space controller with fixed impedance). As introduced in Section 1, the learner must stack the blocks to build a tower. The blocks are initially placed along the centerline of the table, with block A closer to the robot than block B. The exact positions of each block are randomized. We include multiple versions of the task: `StackChoice` (stack the blocks in either order), `StackAB` (stack the red block A on the green block B), and `StackBA` (stack block B on block A). To study ILG-Learn's ability to adapt to a learner's unique capabilities, we further include the `StackChoice-Outwardview` and `StackBA-Outwardview` variants, which simulate observations collected from an object-detector operating from a camera mounted on the same side of the table as the robot arm. Whenever a block is occluded, we mask the corresponding components of the observation space. Figure 1 illustrates the `Stack` tasks; for more details see Appendix E.

The "*grasp A*" landmark comprises the states where the robot gripper fingers are spread approximately the width of the block, and both fingers contact Block A. "*grasp B*" is defined analogously. The "*stack*" landmark comprises the states where the blocks are in contact and on top of each other. During training, a scripted "human" teacher provides **illustrationCount** (10 in our `StackAB` experiment, and 50 in `StackChoice-Outwardview` and `StackBA-Outwardview`) illustrative observations per `requestGoals` call. The scripted teacher always provides the same set of illustrative observations on subsequent `requestGoals` with the same desired vertex and path to extend. The scripted teacher responds to `querySuccess` call perfectly accurately, according to the landmark definitions defined in Appendix E.

For the `StackAB` RCE baseline, we ran experiments with both 100 and 1000 goal examples, all of block A having been stacked on block B. These are drawn from the same distribution as the illustrative observations provided to ILG-Learn for `StackAB`'s "Stack" landmark. Similarly, we ran `StackAB` BC experiments with 100 and 1000 full-length demonstrations gathered using a scripted policy that includes some random variation (see Appendix E for details). Each demonstration is 200 timesteps in length, with the task typically being completed by around 150 timesteps.

**Point Maze.** We use custom layouts of the Point Maze environment from Gymnasium Robotics (de Lazcano et al., 2023; Fu et al., 2020). The agent is a force-actuated point-mass with a 2-dimensional action space and receives 4-dimensional observations that comprise its position and velocity. The agent must navigate from its starting position in the lower-left to the goal position in the upper-right. We include three variants of a diagonal maze (`DiagonalMaze3x3`, `DiagonalMaze5x5`, `DiagonalMaze7x7`) that differ in the length of the task. We also include a `DiagonalMaze7x7-Coarse` variant that has low landmark density and two variants of a 4x4 maze that differ only in their associated ILG specifications. Illustrations are shown in Figure 3; the landmark regions are the centers of the rooms that are highlighted in pink.

In each `Maze` variant, the scripted teacher provides ILG-Learn with 10 illustrative observation per landmark. Similar to the `StackAB`, we conduct `DiagonalMaze` experiments in which the RCE and BC baselines 100 goal examples or demos (respectively), as well as experiments in which they receive 1000. The RCE baseline with 100 examples all drawn from the final goal region, and the BC baseline with 100 full-length demonstrations gathered by a scripted policy.

## 5.1 Landmark density and learning objective

To disentangle the importance of allowing the teacher to choose the ILG's landmark density from the importance of multi-path specifications, we first turn our attention to tasks specified with a linear ILG. In the `StackAB` environment (Figure 1), the ILG-Learn learner receives a linear ILG that mandates placing the red block A on the green block B. At the end of each learning interval (100k environment steps, exclusive of steps used during resets) the teacher responds to 30 success/failure queries. We allow ILG-Learn and RCE to train for 10 million environment steps and allow BC to train until the validation loss stops decreasing. Appendix F contains learning curves.

ILG-Learn achieves an average success rate of 0.975 for the `StackAB` task (Table 1). This is markedly higher than the success rates achieved by the RCE baseline (0.050) or the BC baseline (0.059) when provided

Table 1: Success rates (mean over 5 trials). ILG-Learn, RCE, and HRM are trained for a maximum of 10 million environment steps; see Appendix E for experimental details and Appendix F for learning curves.

|  | ILG-Learn (ours) | RCE 100 example | RCE 1000 ex. | BC (MLP) 100 demo | BC (DiffP) 1000 demo | HRM |
|---|---|---|---|---|---|---|
| StackAB | **0.975** | 0.050 | 0.392 | 0.059 | 0.28 | 0.005 |
| DiagonalMaze3x3 | **1.00** | 0.00 | 0.600 | 0.991 | 0.91 | 1.00 |
| DiagonalMaze5x5 | **1.00** | 0.00 | 0.200 | 0.989 | 0.68 | 0.00 |
| DiagonalMaze7x7 | **0.995** | 0.00 | 0.00 | 0.476 | 0.689 | 0.00 |
| DiagonalMaze7x7-Coarse | 0.00 | 0.00 | 0.00 | 0.476 | **0.689** | 0.00 |

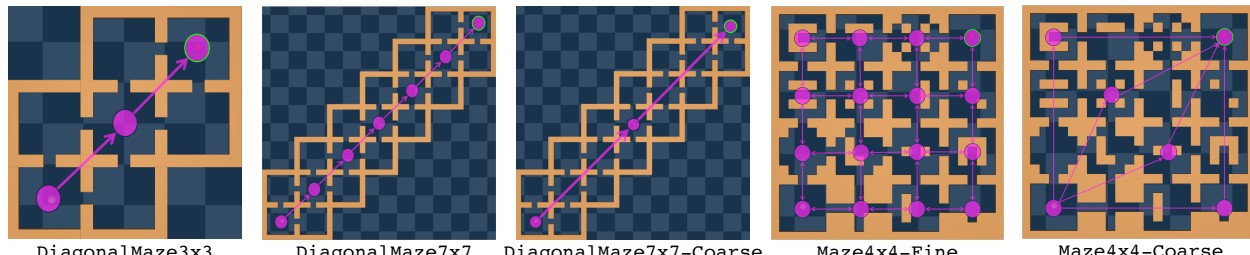

DiagonalMaze3x3    DiagonalMaze7x7    DiagonalMaze7x7-Coarse    Maze4x4-Fine    Maze4x4-Coarse

Figure 3: The `Maze` environments with ILGs superimposed (pink). The agent starts in the lower-left room and must navigate to the upper-right room. `DiagonalMaze5x5` is analogous to the `3x3` and `5x5` variants.

with 100 goal examples or demonstrations, respectively. Even when provided with 1000 examples, RCE only achieves 0.392 average success rate, and BC (with the more modern Diffusion Policy architecture) only achieves 0.28 success rate. The fact that the RCE baseline achieves a low success rate shows that single-frame observations are too sparse to guide the learner towards success; allowing the teacher to specify an ILG with the intermediate "grasp A" landmark was crucial for ILG-Learn's success. On the other hand, the BC baseline, which receives very dense guidance towards the goal, yields policies that rarely succeed, showing that carefully mimicking a teacher's detailed demonstrations does not necessarily yield a good policy.

The `DiagonalMaze` family of environments shows that the teacher's choice of landmark density is more important for tasks with longer horizons. `DiagonalMaze-3x3`, `DiagonalMaze-5x5`, and `DiagonalMaze-7x7` differ only in the length of the path that the learner must travel. Looking again at Table 1, we see that ILG-Learn consistently matches or exceeds the performance of the RCE and BC baselines. Importantly, BC achieves high success rate for the relatively short-horizon `DiagonalMaze3x3` and `DiagonalMaze5x5` but its performance deteriorates dramatically as the size of the maze increases to `7x7`. On the other hand, ILG-Learn consistently yields near-perfect success rates regardless of maze size. This supports the intuition that ILG-Learn can scale to long horizon tasks by (1) allowing the learner flexibility to explore and discover

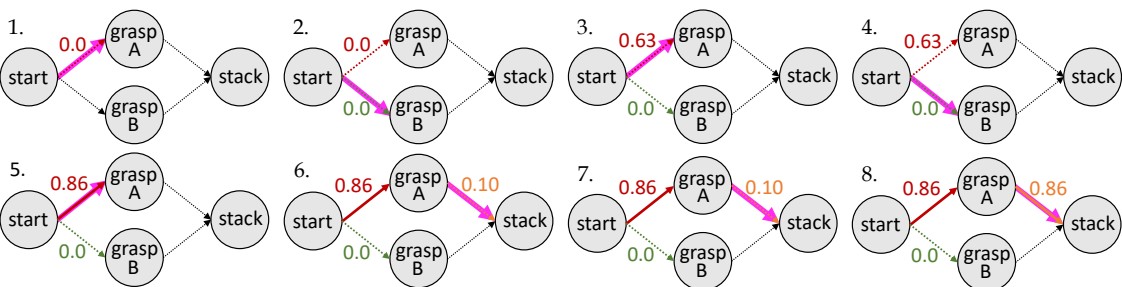

Figure 4: Learning intervals for `StackChoice- Outwardview`. The most recently explored edge is highlighted in pink and results of `estimateSuccess` are annotated.

suitable policies to reach each landmark and (2) incorporating success/failure feedback during training to ensure each edge policy is learned successfully before moving on to train successive policies. This is unlike BC and other imitation learning algorithms that are vulnerable to compounding deviations between the demonstrations and the actual trajectories taken by the agent.

To realize ILG-Learn's benefits, the human teacher must provide an ILG with appropriate landmark density. Given the excessively low landmark density of `DiagonalMaze7x7-Coarse`, ILG-Learn consistently fails to learn a successful policy (while the ILG of `DiagonalMaze7x7` yields near-perfect policies). Such a failure occurs when the exploration needed to learn an edge policy exceeds the capabilities of ILG-Learn's underlying example-based control algorithm. We provide advice for selecting landmark density in Section 6.

To investigate the efficacy of example-based control as ILG-Learn's low-level policy learning algorithm, we compare the performance of ILG-Learn with that of HRM. In `StackAB` and each `DiagonalMaze` environment, we provide HRM with a reward machine in which each reward machine state corresponds to a landmark in our ILG specification. To encourage progress toward completing the long-horizon task, the reward machine provides dense rewards that encourage progress to the next landmark, which triggers a transition to the next reward machine state when reached. The dense reward functions for `StackAB` are directly inspired by the original environmental reward of the original `robosuite Stack` benchmark. For the `DiagonalMaze` tasks, the reward is negative euclidean distance to the center of the next landmark (see Appendix D.4 for details).

We find that for the relatively short-horizon task `DiagonalMaze3x3`, HRM achieves a perfect success rate of 1.00. However, HRM fails to complete the larger `DiagonalMaze` environments. HRM also achieves a low average success rate (0.015) for `StackAB`. While in-depth reward engineering could improve the performance of HRM, the fact that ILG-Learn achieves high success rates with an intuitive ILG specification and 10 illustrative observations per landmark suggests that our approach may be a viable alternative to hierarchical reinforcement learning for long horizon tasks.

## 5.2 Multi-path specifications

A critical attribute of the proposed ILG specification is that it can include multiple source-to-sink paths, each representing an approach to complete the long-horizon task. To evaluate whether ILG-Learn can discover the path that best suits the learner's capabilities, we turn to the `StackChoice-Outwardview` task introduced in Section 1 and illustrated in Figure 1. The task is to stack the blocks (in either order). Importantly, the learner observes the position of each block via a simulated object detector based on a camera situated on the same side of the table as the robot, facing outward toward the blocks. Since the red block A starts closer to the camera than the green block B, the learner cannot detect the position of block B until a block is moved to break the occlusion.

Table 2: Success rate (mean of 5 trials) of ILG-Learn given multi-path specifications.

| StackChoice-OutwardView | StackBA-OutwardView | Maze4x4-Fine | Maze4x4-Coarse |
|---|---|---|---|
| **0.952** | **0.147** | **1.00** | **0.800** |

As expected, in all five trials (random seeds) of ILG-Learn, the learner acquired a policy that follows the path "$start \rightarrow grasp\ A \rightarrow stack$" achieving an average success rate of 0.952. Figure 4 illustrates the exploration process followed by the learner during one representative trial. Initially, the learner explores the ILG by alternating between allocating learning intervals to the ($start, grasp\ A$) edge as well as the ($start, grasp\ B$) edge. However, after 5 training intervals (about 700 thousand total environment steps) the learner successfully acquires a policy for the ($start, grasp\ A$) edge, and is thus able to reach the "$grasp\ A$" landmark. ILG-Learn's best-first search heuristic (detailed in Appendix A) guides further exploration along this path, focusing on the ($grasp\ A, stack$) edge. At the end of the $8^{th}$ learning interval (slightly more than 1 million total environment steps), the learner has acquired a policy that it estimates has a success probability of 0.86. Since this exceeds the user-specified **successThreshold** of 0.8, ILG-Learn terminates. Note that in reality, the success rate of the final policy is 0.947 (estimated from 1000 rollouts)- discrepancy between

the learner's estimate of its own success probability can arise because (1) the learner only uses a modest number (in this case 30, the value of the **estimationQueries** parameter) of success/failure queries and (2) the learner only queries the final timestep of each rollout while we allow success to occur at any point along the trajectory.

The adaptive exploration scaffolded by the ILG described above is crucial for the ILG-Learn learner to successfully complete the task. To illustrate this, we compare against the `StackBA-Outwardview` task, which differs from `StackChoice-Outwardview` only in the ILG provided to the learner. `StackBA-Outwardview` forces the learner to start by grasping block B, which is very hard for the learner, since block B is initially occluded. As expected, ILG-Learn achieves a low average success rate of 0.147 on `StackAB-Outwardview`.

If the human teacher were able to predict which path would be easiest for the learner to follow, the teacher could simply provide a linear ILG (in the the above example, "*start* $\to$ *grasp A* $\to$ *stack*"). However, such upfront prediction may pose a substantial burden to the teacher. A key benefit of using illustrative observations to specify landmarks is that the teacher does not need to be intimately familiar with the learner's perception capabilities—the same specification might even be used for multiple robots with different camera viewpoints. Moreover, using example-based control as a subroutine to let the learner acquire policies to achieve each landmark frees the teacher from having to worry about low-level details of the learner's motor capabilities. To fully realize these freedoms afforded to the teacher, the high-level task specification must also be, to some degree, agnostic to the intricacies of the learner's perception, control, and policy learning capabilities. A branching ILG provides this freedom to the teacher: the teacher can provide an ILG to scaffold exploration and allow the learner to interact with the environment to discover a plan that best fits its unique capabilities.

## 6 Discussion

In Section 5 we observed that ILG-Learn can learn successful policies for manipulation and navigation tasks. In this section, we discuss how the ILG definition affects the feasibility and efficiency of learning, providing both advice for users of ILG-Learn and highlighting directions for future work.

**Human effort.** To apply ILG-Learn, the human teacher must provide an ILG decomposition that contains at least one path that can feasibly be learned via iterated example based control. Although allowing the teacher to provide a branching ILG reduces the teacher's burden to predict the best landmark decomposition, some tasks may be hard for humans to effectively decompose. An important direction for future work is to conduct a careful user study, in the style of Cui et al. (2021); Biyik et al. (2020); Fitzgerald et al. (2023), to investigate the degree of human effort needed to specify an ILG that facilitates successful and efficient application of ILG-Learn. We envision two future directions to reduce the burden of ILG definition: first, we propose to refine the ILG during training by prompting the teacher to provide new intermediate landmarks when the learner struggles. Second, we propose to eliminate the need for manual ILG design altogether, instead allowing the human to specify the task using natural language (and perhaps a few final goal images or demonstrations) and then prompting a foundation model to construct a formal specification (taking inspirations from works such as Lang2LTL (Liu et al., 2023) and SayCan (Ahn et al., 2022)), which can in turn be compiled into an ILG (as in DiRL (Jothimurugan et al., 2021)). These directions are complementary; in the context of text-based sequential decision making, ADaPT (Prasad et al., 2024) has shown that an LLM *planner* can decompose difficult tasks on-the-fly in response to the capabilities of a low level *executor*. We believe that iterative refinement of the ILG by a foundation model would improve the general applicability of ILG-Learn.

We also plan to incorporate foundation models to reduce the human annotation burden imposed by ILG-Learn's `estimateProbability` subroutine. For our experiments, we chose parameters such that the human responds to 30 success/failure queries per 100k training steps; for our `StackChoice-Outwardview` experiment this amounted to an average of 360 active queries per trial. While developing ILG-Learn, we tried to reduce this dependence by using a small amount of human-annotated data to train a binary success classifier for each landmark in a manner inspired by Singh et al. (2019). We found that even with hundreds of labeled positive examples gathered from successful executions of the policy being trained, there was often no threshold on

the output of the classifier neural net that reliably separated success from failure. In the future, we plan to incorporate a foundation-model based success detector (e.g. Goko et al. (2024)) to reduce the human annotation burden.

**Subtask difficulty and learning efficiency.** As established in Section 5, decomposing a long-horizon task into intermediate landmarks and providing a branching ILG that permits multiple paths to success allows ILG-Learn to solve long-horizon tasks. Applying ILG-Learn to a new task thus raises the questions "What is the appropriate landmark density?" and "How much branching should the ILG include?" At a minimum, the teacher should design an ILG that is coarse enough that each landmark is meaningful to the human eye (since the teacher must respond to success/failure queries). Beyond this, we recommend that the teacher err on the side of providing a dense ILG with many landmarks and a large amount of branching. The best-first search pattern introduced in Section 4 and detailed in Appendix A is designed to avoid exploring every edge in the ILG.

To gain empirical insight, we compared the performance and efficiency of ILG-Learn on a 2D navigation task using a "fine" ILG (`Maze4x4-Fine`) versus a "coarse" ILG (`Maze4x4-Coarse`). The tasks are visualized in Figure 3 and learning curves can be found in Appendix F. In this case, either ILG suffices to learn a successful policy: we turn our attention to how much efficiency we would lose by choosing the excessively fine decomposition of `Maze4x4-Fine`. Looking at the learning curves, it takes significantly more (comparing medians, 7x as many) total environment steps to learn a successful policy in `Maze4x4-Fine` than in `Maze4x4-Coarse`. Although this loss of efficiency is regrettable, it is preferable to accidentally providing too coarse an ILG and having learning fail altogether.

Finally, we note the amount of training needed to learn a particular edge task varies between random seeds. To reduce this variability future work could develop example-based control algorithms that are sample efficient and learn more robustly. Additionally, in future work we plan to develop a version of ILG-Learn that leverages dense rewards from a foundation model (taking inspiration from VIP (Ma et al., 2023b) and RoboCLIP (Sontakke et al., 2023)). Foundation-model based low-level policy learning also presents the opportunity to replace the role of illustrative observations with a natural language description of each landmark, which could be combined with rewards from a language-vision foundation model such as LIV (Ma et al., 2023a)) to enable long horizon policy learning with neither reward engineering nor human-provided goal observations.

### Acknowledgments

This work is supported in part by NSF Award 2331783, NSF CAREER Award 2239301, DARPA TIAMAT (HR00112490421), and a gift from AWS AI to Penn Engineering's ASSET Center for Trustworthy AI. The authors would like to thank the TMLR reviewers for their valuable and constructive feedback.

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

## A  Edge selection

The `selectEdge` subroutine governs how ILG-Learn allocates learning intervals while exploring the ILG. As described in Section 4, ILG-Learn starts at the source of the ILG and iteratively extends a tree of best-known policies to the other vertices of the ILG. The key insights that inform the design of `selectEdge` are:

- Training an edge policy $\pi_{(u,v)}$ can only begin once the policy $\pi_u$ is learned (frozen), so it makes sense to stop investing learning intervals along an edge once a desired success probability **successThreshold** is reached. This way, future learning intervals can be allocated to downstream edges.

- Different edges of the ILG may require vastly different numbers of training episodes to acquire a successful policy, so it makes sense to dovetail learning intervals between edges so we do not spend too much effort on a particularly difficult edge.

- Only one feasible path needs to be found, so it makes sense to use a best-first search heuristic to avoid exploring the entire ILG.

Our edge selection algorithm is inspired by the exploration order found in Dijkstra's algorithm. In fact, if we set the parameter **intervalsLimit** to 1, our `selectEdge` subroutine implements an exploration order suitable for Dijkstra's algorithm. When **intervalsLimit** is greater than 1, `selectEdge` incorporates dovetailed exploration with early stopping heuristics to try to reduce the number of training episodes needed to learn a satisfactory policy.

Appendix A.1 describes the high-level structure of `selectEdge` that implements dovetailed exploration and early stopping. Appendix A.2 details the heuristic scoring function that `selectEdge` uses to implement best-first exploration. Appendix B provides advice for selection of relevant parameters. The choice of parameters used for the experiments in Section 5 are detailed in Table 3.

### A.1  Dovetailed exploration

To implement early stopping and dovetailed exploration, ILG-Learn keeps track of these vertex and edge sets:

- *exploredVertices.* These are the vertices to which ILG-Learn has learned and solidified a path policy.

- *learnedEdges.* These are edges for which we have fully learned *and frozen* a policy. We freeze a policy once either (1) the empirical success probability exceeds **successThreshold** or (2) **intervalsLimit** learning intervals have been allocated to the edge.

- *abandonedEdges.* These are edges that we will never invest learning effort into because we have already found a better path to get to all of their successors.

- *frontierEdges.* These are edges leaving the explored tree (that is, leaving *exploredVertices*) that are neither fully "learned" nor "abandoned". At each iteration, we need to choose one edge from this set to invest a learning interval into.

All edge sets are initially empty. The logic for maintaining these sets is shown in Algorithm 2. To reduce notational clutter, we assume that *frontierEdges*, and *learnedEdges*, *abandonedEdges* are global variables that persists across calls to `selectEdge`. We also assume global access to *reachProb* (updated in line 15 of Algorithm 1) as well as the existence of the bookkeeping functions:

- $bestSuccessProb(u,v)$ returns the highest estimated probability found by line 13 of Algorithm 1 during any learning interval allocated to $(u,v)$.

- $intervalsElapsed(u,v)$ returns the number of learning intervals that have been allocated to $(u,v)$

---

**Algorithm 2:** `selectEdge`

---

**Input:**
- ILG structure $(U, E, u_0)$
- Parameters **intervalsLimit** and **successThreshold** (introduced in Section 4)
- Scoring heuristic parameters **extensionPenalty** and **exploitationBonus** (introduced in Appendix A.2)
- Global access to *reachProb* (maintained by Algorithm 1)

**Output:** Edge $(u, v)$ for next learning interval, or $None$ if learning is complete.

$learnedEdges \leftarrow learnedEdges \cup \{(u,v) \in E \mid intervalsElapsed(u,v) \geq \textbf{intervalsLimit}\}$;

$learnedEdges \leftarrow learnedEdges \cup \{(u,v) \in E \mid \ \vee\ bestSuccessProb(u,v) > \textbf{successThreshold}\}$;

$abandonedEdges \leftarrow abandonedEdges \cup \{(u,v) \mid \forall u', u \prec u' \rightarrow reachProb_u < reachProb_{u'}\}$;

$exploredVertices \leftarrow \{u_0\} \cup \{v \in U \mid (\exists u, (u,v) \in learnedEdges) \wedge (\forall u, (u,v) \in E \rightarrow (u,v) \in learnedEdges \cup abandonedEdges)\}$;

$frontierEdges \leftarrow \mathrm{outgoingEdges}(exploredVertices) \setminus learnedEdges \setminus abandonedEdges$;

**return** $argmax_{(u,v) \in frontierEdges} \textit{score}((u,v))$

---

### A.2  Score

The `selectEdge` subroutine (Algorithm 2) uses a heuristic `score` function to implement best-first exploration. We design `score` to balance the following desiderata:

- **Exploitation.** Perform best-first search by extending paths that have low cost.

- **Even exploration.** Dovetail exploration of multiple edges, trying to assign the same amount of training intervals to all extensions of the "sufficiently low cost" paths.

- **Anticipation.** Prefer extending paths that have few edges remaining to a final vertex.

As in Appendix A.1, we will assume global access to *reachProb* and access to the bookkeeping function *intervalsElapsed*. The score of an edge comprises three terms:

$$
\begin{aligned}
\texttt{score}(u,v) = \, & exploitationIncentive(u,v) \\
& - intervalsElapsed(u,v) \\
& - minNumberOfEdgesToAFinalVertex(v) \cdot \textbf{edgeExtensionPenalty}
\end{aligned}
$$

We describe the components as follows:

**Term 1: Exploitation.** The exploitation incentive strongly prioritizes investing training effort in edges that *could be part of the highest probability extension of some path in the tree explored so far.* Concretely, let

$$
bestExtensionProb = \max_{(u,v)\in frontierEdges} reachProb(v)
$$

Now let

$$
exploitationIncentive(u,v) = \begin{cases} \textbf{exploitationBonus} & reachProb(u) \geq bestExtensionProb \\ 0 & \text{otherwise} \end{cases}
$$

wher **exploitationBonus** is some very large number.

**Term 2: Even exploration.** Subtracting *intervalsElapsed*$(u,v)$ softens the best-first search by trying to evenly allocate intervals to promising edges. This term will be much less in magnitude than **exploitationBonus**, so it acts as a tie-breaker within our soft best-first search.

**Term 3: Anticipation.** A high choice of **edgeExtensionPenalty** strongly strongly prioritizes extending paths that are close to reaching a final vertex (in terms of how many edges there are).

## B  Hyperparameter selection

We now provide advice for how to select the ILG-Learn-specifc hyperparameters introduced in Section 4 and Appendix A.2. Table 3 presents our choices for each environment. For details about the hyperparameters used for the RCE in ILG-Learn's `learnPolicy` subroutine, please see Appendix D.

- **illustrationCount**: This should be enough goals so that the underlying example-based control algorithm can efficiently learn a successful policy. To provide empirical insight into how to choose a value of **illustrationCount** we performed multiple experiments for a subset of our environments in which we varied the value of **illustrationCount**, please see Table 4. In the Section 5 we used a value of 10 for all of our environments except for the `outwardView` environments, where we used 50. We note that the choice of **illustrationCount** is tied to the specifics of the underlying edge tasks; we believe that ILGs that include landmarks that admit a diverse set of success observations (e.g. a robotic arm can successfully grasp an object with a wide variety of arm angles and grip positions) will benefit from a relatively large number of success examples. We also note that a straightforward extension to ILG-Learn allows the teacher to increase the number of illustrative observations if the learner struggles to master an edge task.

- **episodeLength**: The fixed time horizon used when training each edge policy must be long enough to allow the example-based control algorithm to explore the environment. The ideal value is usually significantly more than the number of timesteps needed for an expert to complete any given edge task.

- **intervalLength** and **intervalsLimit**: The quantity **intervalLength** × **intervalsLimit** should be enough training steps for example-based control to saturate the success probability of the edge policy for any edge in the ILG. Since the number of learning episodes needed by example based control varies depending on random seed, we recommend over-estimating this quantity. To avoid wasting training episodes on already-good policies, we recommend choosing a relatively low value for **intervalLength**: Since the choice **intervalLength** governs the frequency of teacher-intervention (in the form of responses to success/failure queries), we recommend choosing **intervalLength** to be large enough that there is only a modest amount of teacher interaction, yet low enough that learning episodes are not wasted.

- **successThreshold**: We recommend choosing a value of **successThreshold** slightly lower than the success rate of an optimal policy. Choosing a high **successThreshold** causes ILG-Learn to spend a large amount of learning effort on an edge before exploring deeper in the ILG, which may increase the success rate of the final policy at the cost of more environmental and student-teacher interactions.

- **estimationQueries**: We recommend choosing a number that gives adequate confidence in the success rate of the learned policy. In practice, we have observed that example based control often yields policies that either succeed much more or much less than our desired **successThreshold**, so relatively few *estimationQueries* (e.g. 30) suffice. If there are multiple feasible tasks in the ILG, choosing a high value of **estimationQueries** will improve ILG-Learn's ability to perform best-first search. We chose not to implement a statistically-rigorous version `estimateProbability` since doing so would in general require the teacher to respond to many success//failure queries and was not necessary for good end-to-end performance in practice.

- **exploitationBonus**: We recommend choosing a very high value, so that `score` function implements a "soft" best first seach, in which the second and third terms of the `score` function serve as tie-breakers. To achieve this, one can choose parameters such that **exploitationBonus** > **intervalsLimit** + $diameter(G)$ × **edgeExtensionPenalty** where $diameter(G)$ is the diameter of the ILG.

- **edgeExtensionPenalty**: We recommend choosing **edgeExtensionPenalty** to be a guess of how many learning intervals will be required to train an edge along a feasible path to reach the **successThreshold.** Since choosing this parameter requires considerable foresight, we recommend choosing a relatively low value.

Table 3: ILG-Learn-specific parameter selection for the experiments shown in Section 5.

| | DiagonalMaze3x3 | DiagonalMaze5x5 | DiagonalMaze7x7 | DiagonalMaze7x7-Coarse | Maze4x4-Fine | Maze4x4-Coarse | StackAB | StackChoice-Outwardview | StackBA-Outwardview |
|---|---|---|---|---|---|---|---|---|---|
| **illustrationCount** | 10 | 10 | 10 | 10 | 10 | 10 | 10 | 50 | 50 |
| **episodeLength** | 400 | 400 | 400 | 1200 | 200 | 600 | 80 | 80 | 80 |
| **intervalLength** | 100k | 100k | 100k | 100k | 100k | 100k | 100k | 100k | 100k |
| **intervalsLimit** | 100 | 100 | 100 | 100 | 100 | 100 | 100 | 100 | 100 |
| **successThreshold** | 0.8 | 0.8 | 0.8 | 0.8 | 0.8 | 0.8 | 0.8 | 0.8 | 0.8 |
| **estimationQueries** | 30 | 30 | 30 | 30 | 30 | 30 | 30 | 30 | 30 |
| **exploitationBonus** | 101 | 101 | 101 | 101 | 101 | 101 | 101 | 101 | 101 |
| **edgeExtensionPenalty** | 3 | 3 | 3 | 3 | 3 | 3 | 3 | 3 | 3 |

## C    Data dependence

In order to better understand how the performance of ILG-Learn, RCE, and BC depends on the amount of human-provided data, we performed additional experiments in which we varied the amount of illustrative observations, goal examples, or demonstrations provided to each algorithm. We report our findings in Table 4. For ILG-Learn, the choice of ILG-Learn-specific hyperparameters is the same as reported in Table 3 with the exception of **illustrationCount**, which we manipulate.

Table 4: Final success rates (means over 5 trials). Data count denotes the number of illustrative observations per ILG vertex, number of examples for RCE, or number of demonstrations for BC/DiffP. ILG-Learn and RCE are trained for a maximum of 10 million environment steps; see Appendix E for experimental details and Appendix F for learning curves. Entries are omitted where ILG-Learn achieved near-perfect success rate with fewer illustrative observations.

| | ILG-Learn | | | | RCE | | BC (MLP) | | | BC (DiffP) |
|---|---|---|---|---|---|---|---|---|---|---|
| Data count: | 1 | 10 | 50 | 100 | 100 | 1000 | 1 | 10 | 100 | 1000 |
| StackAB | 0.00 | **0.975** | 0.752 | 0.863 | 0.050 | 0.392 | 0.00 | 0.005 | 0.059 | 0.28 |
| DiagMaze3x3 | 1.00 | 1.00 | | | 0.00 | 0.600 | 0.062 | 0.990 | 0.991 | 0.91 |
| DiagMaze5x5 | 0.991 | **1.00** | | | 0.00 | 0.200 | 0.004 | 0.826 | 0.989 | 0.68 |
| DiagMaze7x7 | 0.200 | **0.995** | | | 0.00 | 0.00 | 0.00 | 0.476 | 0.689 | 0.70 |

## D    Implementation details

### D.1    ILG-Learn and RCE

Our implementation is available at `https://github.com/cwatson1998/ilg-learn`. We implement ILG-Learn in Python. We re-implemented RCE (following instructions of Eysenbach et al. (2021)) on top of JaxRL's (Kostrikov, 2022) implementation of SAC (Haarnoja et al., 2017). We use this RCE implementation both as the example-based control subroutine in our ILG-Learn algorithm and as the RCE baseline.

RCE is an actor-critic example-based control algorithm. Rather than train an auxilliary network to provide a loss signal to be used as in reinforcement learning (as is done in earlier example-based control algoritms such as Fu et al. (2018)) the critic directly predicts a time-discounted probability that an observation, action pair being considered by the learner will lead to success. Success is defined as a regression task, in which the user-provided goal examples represent success and a pool of other transitions (in our case, the replay buffer) represent unlabeled data that is unlikely to represent success.

**Hyperparameters.**    We use the same SAC+RCE hyperparameters as Eysenbach et al. (2021), including the SAC-specific hyperparameters that were inherited from Haarnoja et al. (2017), although we increased the width of each hidden layer in the actor and critic MLPs from 128 to 256. The original RCE implementation varied the "n-step returns" and "Q combinator" hyperparameters depending on the particular task. We use 10-step returns and $max$ as the "Q combinator" for all tasks, which matches the RCE authors' choices for their "sawyer-bin-picking" task. We note that the traditional choice of "Q-combinator" would be $min$, following Fujimoto et al. (2018).

Our choice of ILG-Learn-specific parameters (detailed in Appendix B) is shown in Table 3.

**Policy sequencing details.**    Underlying ILG-Learn's compositional approach to long-horizon policy learning is the `sequencePolicies` subroutine that sequences edge policies to form a path policy. As described in Section 4, each edge policy is executed for a fixed horizon (governed by the **episodeLength** parameter). To formalize the behavior of `sequencePolicies` it is convenient to associate each (path or edge) policy $\pi$ with a $horizon$ denoted $horizon_\pi$). We assume that the array $horizon$ of such bookkeeping variables is in global scope and define the `sequencePolicies` subroutine in Algorithm 3.

---
**Algorithm 3:** `sequencePolicies`

---
**Input:**
- Path policy $\pi_u$ and edge policy $\pi_{(u,v)}$
- Parameter **episodeHorizon**
- Mutable bookkeeping dict *horizon*

**Output:** Path policy to reach $v$

**Function** $\pi_v(\tau)$**:**

    **if** $|\tau| \leq horizon_{\pi_u}$ **then**

        **return** $\pi_u(\tau)$;

    **else**

        **return** $\pi_{(u,v)}(\tau)$;

$horizon_{\pi_v} \leftarrow horizon_{\pi_{(u)}} + horizon_{\pi_{(u,v)}}$;

**return** $\pi_v$

---

If *horizon* is maintained such that $horizon_{\pi_{(u,v)}}$ is **episodeLength** for each edge policy $\pi_{(u,v)}$ then Algorithm 3 matches the textual description of Section 4. In our implementation, we noticed that running each edge policy for the fixed horizon resulted in non-smooth motion, because the agent would hesitate after having achieved each landmark (waiting for the associated edge policy's horizon to be fully consumed) before moving on to the next. To reduce this "waiting time" we slightly enriched the information provided by the teacher: every time the learner uses `querySuccess`$(v)$ at the end of an episode to assess the success rate of an edge policy $\pi_{(u,v)}$, if the teacher responds "Success" they also provide the index of timestep at which the agent entered $\beta v$. Then we update $horizon_{\pi_{(u,v)}}$ to be the max of these success timestep indices, plus an addition 15 timesteps of slack term. The results in Section 5 were collected using the above procedure, although we do not believe that the details of our `sequencePolicy` heuristic is an important aspect of ILG-Learn.

**Success estimation details.** As described in Section 4, the `estimateProbability` estimates the probability that a learned edge policy $\pi_{(u,v)}$ successfully reaches $\beta v$. Since we start each rollout of $\pi_{(u,v)}$ from the state distribution induced by executing the path policy $\pi_u$, we naturally obtain an empirical estimate of the probability that the path policy obtained by sequencing $\pi_u$ and $\pi_{(u,v)}$. We call attention to this because our "Dijkstra-style" planning implemented by `selectEdge` only receives the costs of *paths*, not individual edges. If we assume that each edge has an intrinsic fixed cost, then this is an inconsequential bookkeeping detail. However, since $\pi_{(u,v)}$ could reach $\beta(v)$ even if not started from $\beta(u)$ (and moreover because our cost estimates are empirical) it is possible for the estimated cost of a path to be *less* than that of one of its prefixes. Such ill-behaved path probabilities only pose a problem for ILG-Learn if a high-cost prefix dissuades `selectEdge` from exploring what would become the lowest-cost source to sink path. Such a situation is only likely to arise if the teacher provides an ILG that contains landmarks that are both (1) hard to reach and (2) not necessary to scaffold exploration towards subsequent landmarks.

## D.2 BC (MLP) baseline

We implemented the BC baseline in Python using Jax (Bradbury et al., 2018) and Flax (Heek et al., 2024). For each task, we pooled the (observation, next action) pairs from all the demonstrations and trained an MLP to predict the next action given the current observation. Our MLP had 2 hidden layers (each of width 512), we used the Huber loss and a learning rate of 0.0001. For each task, we had a heldout set of 10 validation demonstrations. We stopped training when the validation loss stopped decreasing.

## D.3 Diffusion Policy baseline

We use the implementation of the U-Net based Diffusion Policy as described in (Chi et al., 2023) using PyTorch. We train the policy on 1000 expert trajectory demos for the stack and diagonal point-maze tasks. We optimize for 100 epochs on the point maze tasks and 300 epochs for the stack task and report the mean success rates across different seeds (3 for stack and 5 for point-maze tasks) using checkpoints that achieves the lowest action reconstruction loss. We report the other common hyperparameters used for training in Table 5.

## D.4 HRM baseline

We baseline against the official implementation of *hierarchical reinforcement learning with reward machines.* Toro Icarte et al. (2022). A reward machine is a deterministic finite automaton whose alphabet is a set of *propositions* over environment's state space. In our experiments, the propositions are the names of

| parameter | value |
|---|---|
| prediction horizon | 8 |
| learning rate | 3e-4 |
| weight decay | 1e-4 |
| input embed dim | 256 |
| step embed dim | 256 |
| U-Net downsample dims | (256, 512, 1024) |
| kernel size | 5 |
| num diffusion steps | 100 |
| EMA power | 0.75 |
| batch size | 128 |

Table 5: Diffusion Policy common training hyperparameters

the *landmarks*, defined formally in Appendix E. At each timestep, the current reward machine state determines which reward function is used to supply rewards to the agent. Even though the reward functions are conditioned on the current state of the reward machine, options correspond to transitions between states. In order to encourage an intra-option policy to reach its specified reward machine state there is a tunable bonus hyperparamater (that we set to 1000) that is recieved by the intra-option policy for a successful transition. For our `StackAB`, `StackChoice-OutwardView`, and `DiagonalMaze3x3` we define the reward machines as shown in Figure 5.

In the `StackAB` environment, the reward functions $r_0^A$ is designed to encourage reaching and grasping block A, and $r_1^B$ is designed to encourage completing the stack. The final state has $r_2^A$ which gives a constant reward of 1000. The other `Stack` environments' reward machines are defined analogously, although $r_0^C$ for `StackChoice` encourages reaching either Block A or Block B.

The `DiagonalMaze` family of environments have reward functions that encourage reaching the next landmark (room center). Each reward machine is defined analogously to the `DiagonalMaze3x3` reward machine. The `DiagonalMaze7x7-Coarse` reward machine has only three states, analogous to the three vertices of the `DiagonalMaze7x7-Coarse` ILG.

We define the reward functions as follows:

$$r_0^A(s, a, s') = (1 - \tanh(10 \, dist(s', \text{BlockA})) * 0.25 + 0.25[grasp(s', \text{BlockA})]$$
$$r_0^B(s, a, s') = (1 - \tanh(10 \, dist(s', \text{BlockA})) * 0.25 + 0.25[grasp(s', \text{BlockA})]$$
$$r_0^C(s, a, s') = \max(r_0^A(s, a, s'), r_0^B(s, a, s'))$$
$$r_1^A(s, a, s') = \max([lifted(BlockA)](1 + 0.5(1 - \tanh(horizontalDist(s', \text{BlockA})))), r_0^A(s, a, s'))$$
$$r_2^B(s, a, s') = \max([lifted(\text{BlockB})](1 + 0.5(1 - \tanh(horizontalDist(s', \text{BlockB})))), r_0^A(s, a, s'))$$
$$r_0^D(s, a, s') = - \, dist(s', \text{nextRoomCenter})$$

where in the above definitions, $dist(s', \text{BlockA})$ denotes Euclidean distance between the center of the agent's end effector and the center of block A. We also use the notation *dist* for the analogous concept in the `DiagonalMaze`. Similarly, *horizontalDist* calculates Euclidean distance, after both points have been projected onto the horizontal plane. All final state rewards are $r(s, a, s') = 1000$. This design pattern is adopted from the original HRM paper. The preceding rewards for the `Stack` family of tasks are heavily inspired by the original reward signal from `robosuite`.

Internally, HRM uses DQN to select options, and DDPG as the intra-option control policy. We use the same hyperparameters that the original authors used for their MuJoCo-based experiments.

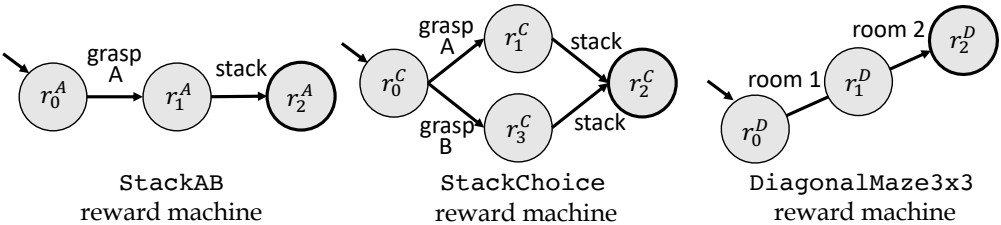

Figure 5: Reward machines structure for HRM baseline.

# E   Environment Details

Our manipulation environment is built in `robosuite` (Zhu et al., 2020). Our maze environment is built in de Lazcano et al. (2023) and is a customized of the Maze2D environment originally introduced in D4RL (Fu et al., 2020). All our experiments use Gymnasium Towers et al. (2023) and MuJoCo Todorov et al. (2012).

## E.1   Stack

We adapt the "Stack" environment that is included in `robosuite`. This environment simulates a 7-DoF Franka Panda robot arm. We use the provided 7-dimensional continuous action space that includes an operational space controller with fixed impedance. We also use the provided observation space, which is 55-dimensional and comprises the robot's joint angles, joint velocities, end effector position, end effector quaternion, gripper position, gripper velocity, the position of each block, the quaternion of each block, the gripper-object distance of each block, and the distance between the two blocks.

We make the following modfications to the original environment:

1. The side length of each cubical block is reduced from 0.05 meters to 0.04m. We do this so we can include more randomness in the initial block placements.

2. The two blocks always initialize along the centerline (parallel to the $x$-axis in the simulation, which is *depth* from the perspective of the robot) of the table with block A closer to the robot than block B. The exact positions are determined randomly in each reset as follows: we uniformly select two $x$ coordinate values in the range $(-0.2, 0.2)$. If the values are at least 0.05 meters apart, we let block A start at the lower $x$ value and block B start at the greater $x$ value. Otherwise, we repeat this process until we obtain sufficiently spaced $x$ values.

3. For **StackChoice-Outwardview** we allow the blocks to be stacked in either order; for **StackBA-Outwardview** we require block B to be stacked on block A.

4. For **StackChoice-Outwardview** and **StackBA-Outwardview** we implement simulated occlusion, as if the observations are produced by an object detector that is mounted on the same side of the table as the robot. To do this, we position a MuJoCo camera at the position indicated by the pink camera icon in Figure 1. At each timestep, if *no pixel* belonging to either block A or block B is present in the ground-truth object segmentation yielded by this camera we mask all components of the observation corresponding to that cube with the value $-2$ (which is far from the observation values encountered in normal operation).

**Landmark definition.**   To simulate the teacher's response to the `querySuccess` queries during training, we defined $\beta(grasp\ A)$ to be all states where block A is at least 0.021 meters above the surface of the table (and similarly for $\beta(grasp\ B)$). We defined $\beta(stack)$ analogously to the success condition of the original `robosuite` stacking environment: the cubes must be touching each other, the gripper cannot be touching the block on top of the tower, and the block on top must be both above the surface of the table and aligned horizontally with the other block.

**Examples and demonstrations.** To generate the landmark examples and demonstrations for behavior cloning, we use a handwritten scripted policy. We inject noise into this scripted policy so that it covers a variety of grip locations and arm paths. We also tried using teleoperated demonstrations (and examples gathered therefrom) for the `StackAB` task and did not find significantly different performance so we elected to use scripted policies to generate all data. While gathering demonstrations, we injected a small amount (uniform with magnitude 0.1) of noise into the simulated actions (but not the action labels in the stored demonstration), similar to DART (Laskey et al., 2017). We 10 held-out validation demonstrations to evaluate loss during BC training.

The example observations for the "grasp" landmarks are taken after the arm has grasped *and lifted the block by* ∼0.02 *meters.* We found that slightly lifting the block greatly improves RCE's ability to learn a successful policy, even in the absence of occlusion. We believe this is because grasps that do not lift the block are very close in observation space to "near grasps" that do not make firm contact with the block, yielding a difficult discrimination task for the RCE critic.

### E.2   Maze

We customized Gymnasium Robotics' Maze2D environment; our layouts are shown in Figure 3. The agent is a force-actuated point mass with a 2-dimensional continuous action space. The observation is continuous and 4-dimensional, comprising the agent's current position and velocity, but not the goal location or the location of any of the maze's walls.

**Landmark definition.** Each landmark is the center of a room, as illustrated in Figure 3. Each square room has a side-length of 3.6; the landmark region is a circle of diameter 1 centered within the room. Only the agent's position (not velocity) determines membership in a landmark.

**Examples and demonstrations.** To provide illustrative observations, we include one illustrative observation from the center of the landmark, with the remaining illustrative observations drawn uniformly at random from positions within the landmark. All illustrative observations contain 0 velocity.

The demonstrations are gathered with handwritten scripted policies that include a small amount of noise to better cover the state space. For each task, all demonstrations traverse the same sequence of rooms to reach the final goal. For each environment, we include 10 demonstrations and an additional 5 held-out demonstrations for evaluating loss during training.

## F   Learning curves

We report the overall success rate of the entire task; for ILG-Learn this means that the learner must explore a source-to-sink path before achieving nonzero success rate. Note that some trials of ILG-Learn stop before reaching 10m environment steps, because ILG-Learn allows early stopping. In this case, we extend the success rate of the latest training checkpoint for the rest of the 10m steps (even though no further training occurs). We mark this extension with dashed line.

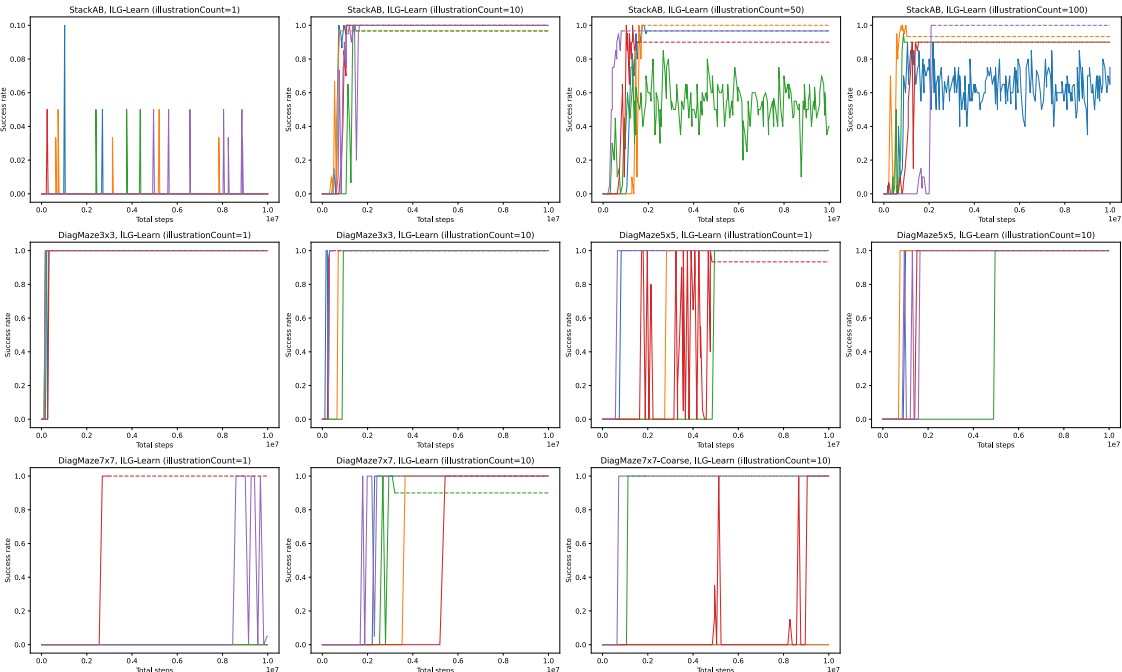

Figure 6: Detailed learning curves for ILG-Learn. Each line represents one of 5 random seeds.

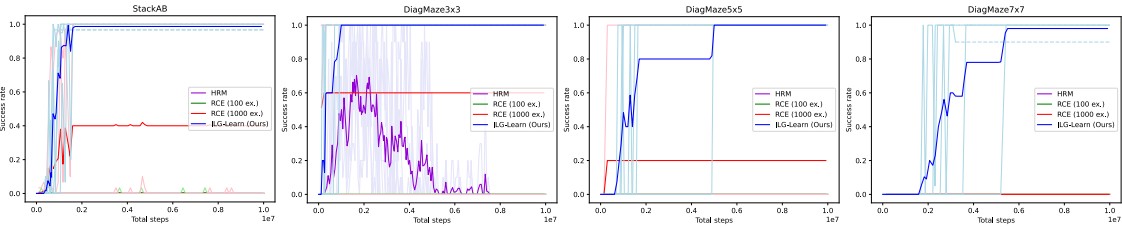

Figure 7: Learning curves showing success rate over total environment steps for ILG-Learn, RCE, and HRM baseline experiments from Table 1. The light lines show individual random seeds, the dark lines show the mean of 5 random seeds.

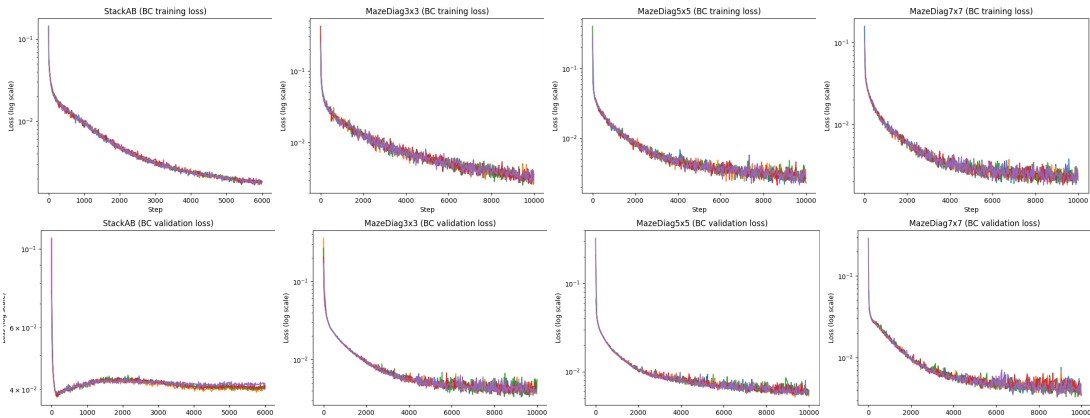

Figure 8: Learning curves showing Huber loss during offline training for the BC (MLP) experiments of Section 5. Each line represents one of 5 random seeds. We chose a number of training steps that was sufficient for validation loss to stop decreasing. For `StackAB` we found this to occur at about 6000 steps, for the other environments we found this to occur at about 10000 steps.

