# OpenReview forum: "Illustrated Landmark Graphs for Long-horizon Policy Learning"
_TMLR — Accepted by TMLR_

### Review · Reviewer_3rxy · 2024-12-13

**Summary Of Contributions:**

The paper proposes landmark graphs to guide and compartmentalize the demonstration learning process that is guided by examples instead of full-length trajectories. The main advantage is that the definition of landmarks is driven by a human, which makes it easy to provide demonstrations for the landmarks. Policies are learned for each edge of the graph so that the complexity of each subtask is reduced and then combined with a search-like algorithm to obtain a controller that fulfills the entire task. The proposed approach is evaluated on a grasping task and a maze to study landmark density and the effect of different sensor perspectives when providing the examples.

The paper is mostly well-written.

**Audience:**

Yes

**Claims And Evidence:**

Yes

**Requested Changes:**

- The abstract does not clearly explain the method proposed, e.g., what is the relation to finite state automata [1] - what are ILGs? what are illustrative observations?
- The introduction does not clarify the contributions enough, e.g., the difference to finite state automata should be clear, it should be clearer how your approach differs from hierarchical learning and TAMP [3]
- Fig. 1 can be improved. Especially the right column is not easy to understand - graphs of tasks and graphs for other approaches. Possibly split the Figure in one to explain the tasks regarded and one to illustrate how your approach compares to others.
- Fig 2. was not really helpful for my understanding and in my opinion, also other methods live in this yellow space [3] [4]
- The related literature misses a comparison to task and motion planning [3] and also the similarity to automata-guided learning [1,2] should be clearer
- In my view, your approach uses the human to abstract the problem into multiple subtasks that are hopefully easier to learn. Another approach would be specifying the tasks with temporal logic formula and synthesizing the graph, aka automata, for that, which would be more automated and less human hand-engineering driven. Maybe you can comment on this perspective in the related work.
- It would be useful to relate the ILG to automata literature in Sec. 3: IL = states, landmark map = labeling function
- Sec. 5 would be more clear if the experimental setup and results were more clearly separated in subsection
- It is not clear to me why the chosen baselines RCE and BC are a good choice. You are motivated by being in the gap between imitation learning and reinforcement learning - why not choose an RL algorithm with sparse rewards from a reward machine?


Minor:
- There seem to be typos on page 3 "bahavior cloning" page 11 "StackAB-Inwardview" and page 12 "human err"
- The style guide specifies that the formatting of Algorithms and Figures should use the full column
- I was wondering if your approach is extendable to tasks with dynamic obstacles, e.g., autonomous driving. This seems to be not the case since the ILG would likely change depending on a specific traffic scenario


References:

[1] Beyazit Yalcinkaya, Niklas Lauffer, Marcell Vazquez-Chanlatte, and Sanjit A. Seshia, “Compositional Automata Embeddings for Goal-Conditioned Reinforcement Learning,” to appear in NeurIPS, 2024.

[2] Learning minimal automata with recurrent neural networks BK Aichernig, S König, C Mateis, A Pferscher, M Tappler
Software and Systems Modeling, 1-31

[3] Curtis, A., Fang, X., Kaelbling, L.P., Lozano-Pérez, T. and Garrett, C.R., 2022, May. Long-horizon manipulation of unknown objects via task and motion planning with estimated affordances. In 2022 International Conference on Robotics and Automation (ICRA) (pp. 1940-1946). IEEE.

[4] LTL and Beyond: Formal Languages for Reward Function Specification in Reinforcement Learning. Alberto Camacho, Rodrigo Toro Icarte, Toryn Q. Klassen, Richard Valenzano, Sheila A. McIlraith. Proceedings of the Twenty-Eighth International Joint Conference on Artificial Intelligence Understanding Intelligence and Human-level AI in the New Machine Learning era. Pages 6065-6073. https://doi.org/10.24963/ijcai.2019/840

**Strengths And Weaknesses:**

Strengths:
- interesting idea to combine an abstract human-designed interpretation of the task (i.e., IL graph) with learning from demonstrations and seems especially relevant for long-horizon tasks
- solid evaluation on a robotic task and a 2D maze

Weaknesses:
- the method and contribution are not clear until Sec. 4
- the similarity to formal methods guided learning is not clear enough from the related work

---

> ### Author Response · Authors · 2025-02-06
> **Response to Reviewer 3rxy (part 1)**
>
> We thank the reviewer for their detailed review.
>
> **The method and contribution are not clear until Sec. 4**
>
> We thank the reviewer for this comment. We have revised the abstract and introduction to improve the clarity of the proposed specification format (ILG) and learning method (ILG-Learn). Furthermore, we have updated the list of contributions to clearly highlight the benefits of our proposed method.
>
> **the similarity to formal methods guided learning is not clear enough from the related work**
>
> We have added a paragraph to the related works section (Section 2) that explains more clearly the connection between the ILG, temporal logic specifications, and reward machines [5].
>
> To draw a direct comparison with temporal logic specifications we note that the ILG-definable specifications correspond to the following fragment of the SPECTRL [6] temporal logic:
>
>                                         φ ::=  achieve atomic_proposition | φ ; φ | φ or φ
>
> that excludes the “ensuring” construct. Concerning reward machines, both an ILG and a reward machine expose the symbolic structure of a task to a learner, however they do so in different ways. An ILG exposes the high-level objective (reach a final landmark) to the learner, who must combine graph-based planning and example-based control to obtain local guidance. On the other hand, a reward machine immediately exposes local guidance to the learner (in the form of the initial reward machine state's reward function); the learner may leverage the reward machine's symbolic structure to guide exploration as it seeks to maximize accumulated reward, which may or may not require reaching a particular reward machine state.
>
> **The abstract does not clearly explain the method proposed, e.g., what is the relation to finite state automata - what are ILGs? What are illustrative observations?**
>
> We revised our abstract to make the description of ILG and ILG-Learn clearer. Regarding the relation to formal-methods based approaches, we revised the introduction (Section 1) to say that the ILG is “inspired by the use of temporal logic to structure reinforcement learning” and that each landmark is “analogous to an atomic proposition of a temporal logic specification.” This comparison is expanded on in the related works (Section 2).
>
> **The introduction does not clarify the contributions enough, e.g., the difference to finite state automata should be clear, it should be clearer how your approach differs from hierarchical learning and TAMP [3]**
>
> We thank the reviewer for this suggestion. We have added a statement to the introduction (Section 1) describing the similarity to specification-guided reinforcement learning and expanded this comparison to encompass automata-based methods in the related works section (Section 2). We also added a sentence drawing a parallel to hierarchical reinforcement learning in the introduction: “...This allows the learner to benefit from temporal abstraction, as in hierarchical reinforcement learning [6] without reward engineering.” Similarly we add a comparison with TAMP to the introduction:“...In this way, the ILG differs from the symbolic abstractions used for Task & Motion Planning, [7] in which the human must define which transitions are feasible before planning begins.”
>
> **It is not clear to me why the chosen baselines RCE and BC are a good choice. You are motivated by being in the gap between imitation learning and reinforcement learning - why not choose an RL algorithm with sparse rewards from a reward machine?**
>
> We thank the reviewer for this suggestion and have revised the manuscript to include Hierarchical Reinforcement Learning for Reward Machines (HRM) [5] as an additional baseline. We observed that sparse rewards are insufficient for HRM to successfully complete even our simplest DiagonalMaze3x3 environment, so we used a reward machine that provides dense rewards for our experiments. Our dense rewards were inspired by the reward machines used in the original HRM paper and the original reward of the Robosuite “Stack” benchmark. This enabled HRM to achieve a high success rate in DiagonalMaze3x3. Although we are still running some HRM baseline experiments due to initial technical difficulties, preliminary results (rollouts logged while while training an HRM agent) suggest that HRM can complete the initial “grasp” part of both the StackAB task and StackChoice-OutwardView, but cannot complete the Stack. If our final evaluation confirms this negative result, that will help show that exposing the structure of a task using a reward machine does not alleviate the need for careful reward engineering and lends evidence that ILG-Learn may be a promising alternative to manual reward specification for long horizon tasks.

---

> > ### Author Response · Authors · 2025-02-06
> > **Response to Reviewer 3rxy (part 2)**
> >
> > **Fig. 1 can be improved. Especially the right column is not easy to understand - graphs of tasks and graphs for other approaches. Possibly split the Figure in one to explain the tasks regarded and one to illustrate how your approach compares to others.**
> >
> > We agree that trying to represent example-based control and imitation learning in the same format as an ILG obscures the distinction between the techniques. We have updated Figure 1 to remove these non-ILG graphs. Rather than introduce an additional figure, we updated the text of section 1 to explain example based control in the context of ILG-Learn. In particular, we write “...to train an edge policy for Start -> Grasp B, the teacher would provide illustrative observations by manipulating the gripper to grasp block B, then letting the learner observe the grasp with its onboard sensors. Example-based control lets the learner use such illustrative observations in lieu of a reward function as it interacts with the environment to learn a control policy to grasp block B.” We believe that this concrete description of example-based prepares the reader for the last paragraph of the introduction (Section 1), where we contrast the density of guidance available in example based control, ILG-Learn, and imitation learning.
> >
> > **Fig 2. was not really helpful for my understanding and in my opinion, also other methods live in this yellow space**
> >
> > Thank you for this comment. We have decided to removed Figure 2, since the schematic diagram does not provide intuition beyond what we convey in the text. Moreover, a two dimensional graph is not expressive enough to accurately characterize the rich landscape of approaches.
> >
> > **In my view, your approach uses the human to abstract the problem into multiple subtasks that are hopefully easier to learn. Another approach would be specifying the tasks with temporal logic formula and synthesizing the graph, aka automata, for that, which would be more automated and less human hand-engineering driven. Maybe you can comment on this perspective in the related work**
> >
> > We thank the reviewer for this comment. We have added a note in the related works section (Section 2) that the ILG-definable specifications are subsumed by LTL that “ILG-Learn can equivalently be seen as a specification-guided reinforcement learning algorithm in which the learner must infer the semantics of each atomic predicate (landmark) from a set of illustrative observations and active success/failure queries.”
> >
> > **it would be useful to relate the ILG to automata literature in Sec. 3: IL = states, landmark map = labeling function**
> >
> > We thank the reviewer for this suggestion, however we observe that our definition of ILG satisfaction differs from the notion of acceptance common in the automata literature. In particular, a trajectory satisfies an ILG so long as any state in the trajectory belongs to a final landmark. In this regard, any ILG corresponds to a two state DFA with an absorbing final state. While the ILG-Learn algorithm is well-suited for an alternate notion of task graph satisfaction that requires reaching each intermediate landmark along a path, we prefer our simpler notion of success because it allows us to focus on the ILG as a scaffold for teacher-learner interaction (that can be compared to imitation learning and example based control) rather than to specify non-Markovian objectives. To preserve this framing, we elected to not draw an explicit comparison to automata in Section 3.
> >
> > **I was wondering if your approach is extendable to tasks with dynamic obstacles, e.g., autonomous driving. This seems to be not the case since the ILG would likely change depending on a specific traffic scenario**
> >
> > ILG-Learn can handle dynamic obstacles within the level of the low-level edge policies. However, we perform planning through the ILG at train time and fix the best path through the task graph. A future extension to ILG-Learn could train a hierarchical policy to dynamically select a path in the ILG, although this would increase the training burden as the learner explores multiple paths to success.
> >
> > **Sec. 5 would be more clear if the experimental setup and results were more clearly separated in subsection**
> >
> > We believe that since each experiment setup addresses a different research question, it is most clear to follow each experimental setup with its results.
> >
> > **There seem to be typos on page 3 "bahavior cloning" page 11 "StackAB-Inwardview" and page 12 "human err"**
> >
> > Thank you; we have fixed these.
> >
> > **The style guide specifies that the formatting of Algorithms and Figures should use the full column**
> >
> > Thank you for directing our attention to this; we revised Table 1, our figures, and Algorithm 1 so that they use the full column.

---

> > > ### Author Response · Authors · 2025-02-06
> > > **Response to Reviewer 3rxy (part 3)**
> > >
> > > ### References
> > >
> > > [1] Beyazit Yalcinkaya, Niklas Lauffer, Marcell Vazquez-Chanlatte, and Sanjit A. Seshia, “Compositional Automata Embeddings for Goal-Conditioned Reinforcement Learning,” NeurIPS, 2024.
> > >
> > > [2] Learning minimal automata with recurrent neural networks BK Aichernig, S König, C Mateis, A Pferscher, M Tappler Software and Systems Modeling, 1-31
> > >
> > > [3] Curtis, A., Fang, X., Kaelbling, L.P., Lozano-Pérez, T. and Garrett, C.R., 2022, May. Long-horizon manipulation of unknown objects via task and motion planning with estimated affordances. In 2022 International Conference on Robotics and Automation (ICRA) (pp. 1940-1946). IEEE.
> > >
> > > [4] LTL and Beyond: Formal Languages for Reward Function Specification in Reinforcement Learning. Alberto Camacho, Rodrigo Toro Icarte, Toryn Q. Klassen, Richard Valenzano, Sheila A. McIlraith. Proceedings of the Twenty-Eighth International Joint Conference on Artificial Intelligence Understanding Intelligence and Human-level AI in the New Machine Learning era. Pages 6065-6073. https://doi.org/10.24963/ijcai.2019/840
> > >
> > > [5] Rodrigo Toro Icarte, Toryn Q. Klassen, Richard Valenzano, and Sheila A. McIlraith. 2022. Reward Machines: Exploiting Reward Function Structure in Reinforcement Learning. J. Artif. Int. Res. 73 (May 2022). https://doi.org/10.1613/jair.1.12440
> > >
> > > [6] Kishor Jothimurugan, Rajeev Alur, and Osbert Bastani. 2019. A composable specification language for reinforcement learning tasks. Proceedings of the 33rd International Conference on Neural Information Processing Systems. Curran Associates Inc., Red Hook, NY, USA, Article 1168, 13041–13051.
> > >
> > > [7] Hutsebaut-Buysse, M., Mets, K., & Latré, S. (2022). Hierarchical Reinforcement Learning: A Survey and Open Research Challenges. Machine Learning and Knowledge Extraction, 4(1), 172-221. https://doi.org/10.3390/make4010009
> > >
> > > [8] Z. Zhao et al., "A Survey of Optimization-Based Task and Motion Planning: From Classical to Learning Approaches," in IEEE/ASME Transactions on Mechatronics, doi: 10.1109/TMECH.2024.3452509.

---

> > > > ### Comment · Reviewer_3rxy · 2025-02-09
> > > > **Reply to reviewer response**
> > > >
> > > > Thanks for your answers and revised version. The additional experiments and revised figures are a significant improvement.
> > > >
> > > > Could you please further clarify the following:
> > > >
> > > > - The specification-guided RL section should clearly convey the main advantage of more formal approaches: no/significantly less human effort. This is currently not clear.
> > > > - Are the HRM baseline results final? The last row in Tab. 1 seems to be incomplete or wrongly highlighted.
> > > > - While I appreciate your comparison with (Toro Icarte et al., 2022) it would be good to discuss more recent approaches like [1] in the human effort section. Why do you prefer that an LLM creates the ILG instead of a more formal synthesis from (potentially LLM-created) temporal logic specifications?
> > > > - Your argument for dynamic environments is not clear to me. I still do not see how your approach would be valuable for such environments. Could you provide a specific example?
> > > > - Please introduce the abbreviation HRM in the main text.

---

> > > > > ### Author Response · Authors · 2025-02-13
> > > > > **Further clarifications**
> > > > >
> > > > > **> The specification-guided RL section should clearly convey the main advantage of more formal approaches**
> > > > >
> > > > > We thank the reviewer for this suggestion, and have added the sentence “The main benefit of specification-guided reinforcement learning is that some (or all) reward engineering effort can be replaced with logical specification, which is usually more intuitive and less error prone” to Section 2.
> > > > >
> > > > > **> Are the HRM baseline results final? The last row in Tab. 1 seems to be incomplete or wrongly highlighted.**
> > > > >
> > > > > We thank the reviewer for pointing this out. We had not filled in results for HRM on the "Coarse" version of the DiagonalMaze7x7 maze task. We have now filled this in, and updated the HRM details in Appendix D.4 to describe the reward machine used for this experiment.
> > > > >
> > > > > **> it would be good to discuss more recent approaches like [1] in the human effort section. Why do you prefer that an LLM creates the ILG instead of a more formal synthesis from (potentially LLM-created) temporal logic specifications?**
> > > > >
> > > > > We agree that it would be best to use an LLM to first create a temporal logic specification; we  have updated the sentence in Section 6 to read: “...we propose to eliminate the need for manual ILG design altogether, instead allowing the human to specify the task using natural language (and perhaps a few final goal images or demonstrations) and then prompting a foundation model to construct a formal specification (taking inspirations from works such as Lang2LTL [2] and SayCan [3]),  which can in turn be compiled into an ILG (as in DiRL [4]).”
> > > > >
> > > > > We appreciate the suggestion to comment on more recent specification and automaton-based approaches to RL. We decided to add a comparison to [1] in Section 2 rather than the human effort subsection of our Discussion. We added the sentence: “ILG-Learn and DiRL both use the specification's structure to scaffold train-time planning; this contrasts with the approach of [1], which uses a neural network to produce an embedding of the specification that serves as input to a goal-conditioned policy.”
> > > > >
> > > > > **> Your argument for dynamic environments is not clear to me. I still do not see how your approach would be valuable for such environments. Could you provide a specific example?**
> > > > >
> > > > > With respect to dynamic environments, ILG-Learn does not add to the adaptability of the underlying example-based control algorithm. Let us consider an idealized autonomous driving scenario with the ILG specification:
> > > > >
> > > > >     Start -> Waypoint 1
> > > > >     Start -> Waypoint 2
> > > > >     Waypoint 1 -> Goal
> > > > >     Waypoint 2 -> Goal
> > > > >
> > > > > Imagine that at the start of each episode there is a mobile obstacle between the agent and either Waypoint 1 or Waypoint 2 (and this can be observed from the agent’s initial position).
> > > > >
> > > > > _ILG-Learn cannot plan dynamically._ Over the course of training, ILG-Learn solidifies a single, fixed path through the ILG. This means that ILG-Learn cannot learn a policy that reactively chooses to visit Waypoint 1 or Waypoint 2 depending on the initial location of the obstacle.
> > > > >
> > > > > _However, ILG-Learn’s edge policies are reactive._ Imagine that ILG-learn solidified the "Start -> Waypoint 1 -> Goal" during training. The edge policy for ``Start -> Waypoint 1’’ was learned using example-based control, which might have been able to learn a policy that dynamically avoids the obstacle (if present) while still reaching Waypoint 1. If so, then ILG’s composite path policy will avoid the mobile obstacle.
> > > > >
> > > > > ---
> > > > >
> > > > > ## References:
> > > > > [1] Beyazit Yalcinkaya, Niklas Lauffer, Marcell Vazquez-Chanlatte, and Sanjit A. Seshia, “Compositional Automata Embeddings for Goal-Conditioned Reinforcement Learning,” NeurIPS, 2024.
> > > > >
> > > > > [2] Liu, J.X., Yang, Z., Idrees, I., Liang, S., Schornstein, B., Tellex, S. &amp; Shah, A.. (2023). Grounding Complex Natural Language Commands for Temporal Tasks in Unseen Environments. _Proceedings of The 7th Conference on Robot Learning_, in _Proceedings of Machine Learning Research_ 229:1084-1110 Available from https://proceedings.mlr.press/v229/liu23d.html.
> > > > >
> > > > > [3] Ahn, Michael et al. “Do As I Can, Not As I Say: Grounding Language in Robotic Affordances.” Conference on Robot Learning (2022).
> > > > >
> > > > > [4] Jothimurugan, K., Bansal, S., Bastani, O., & Alur, R. (2021). Compositional Reinforcement Learning from Logical Specifications. ArXiv, abs/2106.13906.

---

> ### Comment · Reviewer_3rxy · 2025-02-18
> **Response to further clarifications**
>
> Thanks for the detailed answers and corrections in your paper. I am still not convinced by your argument for ILG-Learn and autonomous driving. Often, a reference path can be easily found by exploiting the road structure, and the main challenge for the agent is to appropriately react to the highly dynamic environment (i.e., another traffic participant). However, I believe your method adds value in many other application domains. Thanks again for the clarifications.

---

### Review · Reviewer_XYTR · 2024-12-13

**Summary Of Contributions:**

This paper introduces the Illustrated Landmark Graph (ILG) as a novel framework to address long-horizon sequential decision-making tasks. By using landmarks as intermediate states, the ILG effectively decomposes a complex task into manageable subtasks. The proposed ILG-Learn algorithm integrates this task specification into a policy learning framework that combines planning, policy learning, and teacher feedback. The approach is validated empirically on manipulation and navigation tasks.

**Audience:**

Yes

**Claims And Evidence:**

Yes

**Requested Changes:**

1. The policy learning paragraph in Section 4 states that the policy tries to reach states that *look similar* to the human-provided examples. Could the author clarify how the similarity is quantified?

2. Could the author prepare a table for all hyper-parameters? I would like to know the values of intervalLength, episodeLength, successThreshold, intervalsLimit and the rationale of the choices.

3. In the experiments, more details of the RCE baseline should be provided.

4. Section 5.1 says that each learning interval includes $100$k environment steps, while later it says ILG-Learn is trained for $10000$ environment steps. It is confusing. I think the learning interval should be shorter than the total training. Could the author clarify it?

**Strengths And Weaknesses:**

**Strengths**

1. This paper introduces a novel perspective on task decomposition by enabling a balance between abstract declarative guidance and prescriptive demonstrations.

2. The paper is well-structured, with clear motivation, methodology, and experiments.

**Weaknesses**

1. The method is highly motivated by the research gap between RL and imitation learning. However, the experiments omit RL baselines, such as goal-conditioned RL or hierarchical RL, which are directly relevant. Adding these would strengthen the claims.

2. The comparison with behaviour cloning seems unfair due to significantly less samples provided for behavior cloning. It would be better to show the interactions/demonstrations needed by ILG-Learn and BC to reach a given success rate, say $40$%, $60$% and $80$%. Hopefully, ILG-Learn can reach the same success rate with much less samples.


3. Figure 6 highlights substantial performance variability with different random seeds. This raises concerns about the robustness of ILG-Learn and suggests a need for further discussion or solutions to address instability.

---

> ### Author Response · Authors · 2025-02-06
> **Response to Reviewer XYTR (part 1)**
>
> We thank the reviewer for their detailed review.
>
> **The experiments omit RL baselines, such as goal-conditioned RL or hierarchical RL, which are directly relevant. Adding these would strengthen the claims.**
>
> We thank the reviewer for this suggestion. We have revised the manuscript to include Hierarchical Reinforcement Learning for Reward Machines (HRM) [1] as an additional baseline. HRM is an instantiation of the options framework for hierarchical RL that interleaves training the high-level policy over options and the low-level intra-option policies. Moreover, the learner receives structured rewards from a human-specified reward machine that exposes the high-level structure of the task to the learner. In this respect, HRM combines aspects of GCRL: each intra-option policy receives a reward for triggering a transition to its target reward machine state. This structure is similar to the ILG; HRM is unique among our baselines because it is well-suited to tasks such as our StackChoice-Outwardview that require high-level planning. We manually design reward machines that are inspired by the reward machines used in the original HRM paper and the original reward of the Robosuite “Stack” benchmark [2]. This enabled HRM to achieve a high success rate in DiagonalMaze3x3. Although we are still running some HRM baseline experiments due to initial technical difficulties, preliminary results suggest that HRM can complete the initial “grasp” part of both the StackAB task and StackChoice-OutwardView, but cannot complete the Stack. If our final evaluation confirms this negative result, that will help show that exposing the structure of a task using a reward machine does not alleviate the need for careful reward engineering and lends evidence that ILG-Learn may be a promising alternative to manual reward specification for long horizon tasks.
>
> **The comparison with behaviour cloning seems unfair due to significantly less samples provided for behavior cloning. It would be better to show the interactions/demonstrations needed by ILG-Learn and BC to reach a given success rate**
>
> We ran additional experiments to investigate how the number of illustrative observations, goal examples, and demos affect the performance of ILG-Learn, RCE, and BC, respectively and have included the results of the study as Table 4 in the Appendix. We found that increasing the number of demonstrations provided to BC from 10 to 100 yields higher success rates in the StackAB and DiagonalMaze, although even with 1000 demonstrations and a more sophisticated policy architecture (Diffusion Policy [3]). BC was unable to reach more than a 0.28 success rate on StackAB. This strengthens our claim that ILG-Learn is a promising alternative to imitation learning for long horizon tasks, since ILG-Learn was able to achieve a 0.76 success rate with a mere 10 illustrative observations per landmark.
>
> **Figure 6 highlights substantial performance variability with different random seeds.**
>
> The performance variability stems from the lack of robustness of the underlying policy learning subroutine, which in our implementation is the RCE algorithm. We observe that our proposed framework is agnostic to the method used to learn these low-level control policies. We believe future works could improve the robustness of our framework by (a) augmenting the learning process with dense feedback signals from Foundation Models (e.g. VIP [4] or RoboCLIP [5]) to reliably learn the required control policies, and (b) explore alternate example-based learning methods [6,7]. We have updated our discussion section (Section 6) to identify this direction.
>
> **The policy learning paragraph in Section 4 states that the policy tries to reach states that look similar to the human-provided examples. Could the author clarify how the similarity is quantified?**
>
> We realize now that this description is vague and have reworded it to say: “the learner would interact with the environment to learn a policy that tries to reach states that yield observations that look similar to the illustrative observations of the ``Grasp A'' landmark. This similarity is quantified internally by the example-based control algorithm. Our implementation uses RCE [8] in which a critic network is trained to predict a time-discounted success probability for (observation, action) pairs, while regressing the success probability of illustrative observations to 1. The critic's scores are used to optimize an actor policy, as in the soft actor critic RL algorithm [9].

---

> > ### Author Response · Authors · 2025-02-06
> > **Response to Reviewer XYTR (part 2)**
> >
> > **Could the author prepare a table for all hyper-parameters? I would like to know the values of intervalLength, episodeLength, successThreshold, intervalsLimit and the rationale of the choices.**
> >
> > We have included details of all the hyperparameters (Table 2 in Appendix) and the rationale for the choices made are included in Appendix B.
> >
> > **In the experiments, more details of the RCE baseline should be provided.**
> >
> > We have updated Section 5: Experiments to point to our description of RCE in Appendix D.1: Implementation Details.
> >
> > **Section 5.1 says that each learning interval includes 100k environment steps, while later it says ILG-Learn is trained for 10000 environment steps.**
> >
> > Thank you for bringing this to our attention; this is a typographical error and has been fixed in our revised manuscript. Each learning interval is indeed 100k environment steps; we meant to say that ILG-Learn is trained for 10 million environment steps.
> >
> > ---
> >
> > ### References
> >
> > [1] Rodrigo Toro Icarte, Toryn Q. Klassen, Richard Valenzano, and Sheila A. McIlraith. 2022. Reward Machines: Exploiting Reward Function Structure in Reinforcement Learning. J. Artif. Int. Res. 73 (May 2022). https://doi.org/10.1613/jair.1.12440
> >
> > [2] Yuke Zhu and Josiah Wong and Ajay Mandlekar and Roberto Martin-Martin and Abhishek Joshi and Kevin Lin and Soroush Nasiriany and Yifeng Zhu. robosuite: A Modular Simulation Framework and Benchmark for Robot Learning. 2020.
> >
> > [3] Cheng Chi, Zhenjia Xu, Siyuan Feng, Eric Cousineau, Yilun Du, Benjamin Burchfiel, Russ Tedrake, & Shuran Song (2024). Diffusion Policy: Visuomotor Policy Learning via Action Diffusion. The International Journal of Robotics Research
> >
> > [4] Yecheng Jason Ma, Shagun Sodhani, Dinesh Jayaraman, Osbert Bastani, Vikash Kumar, Amy Zhang. VIP: Towards Universal Visual Reward and Representation via Value-Implicit Pre-Training. ICLR. 2023.
> >
> > [5] Sumedh A Sontakke, Jesse Zhang, Sébastien M. R. Arnold, Karl Pertsch, Erdem Bıyık, Dorsa Sadigh, Chelsea Finn, and Laurent Itti. 2023. RoboCLIP: one demonstration is enough to learn robot policies. In Proceedings of the 37th International Conference on Neural Information Processing Systems (NIPS '23). Curran Associates Inc., Red Hook, NY, USA, Article 2430, 55681–55693.
> >
> > [6] Avi Singh, Larry Yang, Kristian Hartikainen, Chelsea Finn, Sergey Levine. End-to-End Robotic Reinforcement Learning without Reward Engineering. Robotics: Science and Systems, 2019.
> >
> > [7] Kevin Li and Abhishek Gupta and Vitchyr H. Pong and Ashwin Reddy and Aurick Zhou and Justin Yu and Sergey Levine. Reinforcement Learning with Bayesian Classifiers: Efficient Skill Learning from Outcome Examples. OpenReview. 2021.
> >
> > [8] Benjamin Eysenbach and Sergey Levine and Ruslan Salakhutdinov. Replacing Rewards with Examples: Example-Based Policy Search via Recursive Classification. NeurIPS (2021).
> >
> > [9] Haarnoja, T., Zhou, A., Abbeel, P. &amp; Levine, S.. (2018). Soft Actor-Critic: Off-Policy Maximum Entropy Deep Reinforcement Learning with a Stochastic Actor. Proceedings of the 35th International Conference on Machine Learning, in Proceedings of Machine Learning Research 80:1861-1870 Available from https://proceedings.mlr.press/v80/haarnoja18b.html.

---

> > > ### Comment · Reviewer_XYTR · 2025-02-10
> > >
> > > Thanks for your replies. Some concerns have been addressed, but two remain:
> > >
> > > 1. For **Question 3**, the authors state that the variability stems from the RCE subroutine and also mention that the proposed framework is agnostic to the algorithm used in the subroutine. Is it possible for the authors to try another subroutine algorithm to achieve more robust performance? The variability is too large to be ignored.
> > >
> > > 2. For **Question 4**, the reworded description-states that yield observations that look similar to the illustrative observations-is still unclear about how similarity is quantified. It is not sufficient to say that it is quantified internally by the example-based control algorithm.

---

> > > > ### Author Response · Authors · 2025-02-13
> > > > **Variability and learning curves**
> > > >
> > > > **> Is it possible for the authors to try another subroutine algorithm to achieve more robust performance? The variability is too large to be ignored.**
> > > >
> > > > We thank the reviewer for encouraging us to look deeper into this. While also updating the learning curves in Appendix F to differentiate individual random seeds using color, we noticed that some ILG-Learn experiments were originally run with software settings that disabled the termination feature of ILG-Learn that is described in Section 4 and Appendix A. We re-ran these experiments: the new learning curves for DiagonalMaze3x3 and DiagonalMaze5x5 show considerably less variability between random seeds, since much of the variability in the original trials occurred after ILG-Learn’s termination criteria were met (dashed lines indicate early termination). We believe that the remaining variability between seeds is now typical of any exploration-intensive reinforcement learning-based approach, where chance events play a key role in policy improvement and convergence.
> > > >
> > > > Still, for completeness, we now also report the results of our new experiments with VICE-RAQ [1] as the example-based control subroutine, as the reviewer requested. For our new experiments that use VICE-RAQ as the example-based control subroutine, we focused on the initial "Start -> Grasp A" edge of the StackAB task (with illustrationCount=10). We justify this choice because it showed significant variability in our original experiments: the five random seeds of ILG-Learn (illustrationCount=10) took 3, 4, 5, 7, and 10 learning intervals (each representing 100k environment steps) to learn the subtask. For reference, the most variability exhibited by ILG-Learn (illustrationCount=10) for DiagonalMaze7x7 was on that task’s penultimate edge task, with 1, 1, 2, 3, and 12 learning intervals.
> > > >
> > > > Since VICE-RAQ is known to perform better with more human-provided goal examples, we studied VICE-RAQ’s performance with illustrationCount=100. We ran 5 random seeds. Of these, three learned a policy that achieves at least 0.8 success rate (the value of the successThreshold hyperparameter) for the "Start -> Grasp A’", taking 6, 18, and 22 learning intervals. The remaining 2 random seeds failed to learn a successful edge policy within 50 learning intervals.
> > > > To sum up, we tracked down much of the variability between seeds to a minor oversight in some of our experiments, and have now rectified it, thank you!
> > > >
> > > > **> For Question 4, the reworded description-states that yield observations that look similar to the illustrative observations-is still unclear about how similarity is quantified. It is not sufficient to say that it is quantified internally by the example-based control algorithm.**
> > > >
> > > > We thank the reviewer for encouraging us to explain more clearly. In Section 4 of our updated manuscript we write, “...This similarity is quantified by the example-based control algorithm. Our implementation uses RCE, [2] which is inspired by actor-critic reinforcement learning. In RCE, the critic network is trained to predict a time-discounted success probability for observation-action pairs. Success is defined as a regression task: the success probability of illustrative observations is regressed to 1 while other observations (collected in a replay buffer) are assumed to not represent success. The critic's scores are used to optimize the policy, similarly to the soft actor critic RL algorithm [3].”
> > > >
> > > > ---
> > > >
> > > > ## References:
> > > > [1] Avi Singh, Larry Yang, Kristian Hartikainen, Chelsea Finn, Sergey Levine. End-to-End Robotic Reinforcement Learning without Reward Engineering. Robotics: Science and Systems, 2019.
> > > >
> > > > [2] Benjamin Eysenbach and Sergey Levine and Ruslan Salakhutdinov. Replacing Rewards with Examples: Example-Based Policy Search via Recursive Classification. NeurIPS (2021).
> > > >
> > > > [3] Haarnoja, T., Zhou, A., Abbeel, P. &; Levine, S.. (2018). Soft Actor-Critic: Off-Policy Maximum Entropy Deep Reinforcement Learning with a Stochastic Actor. Proceedings of the 35th International Conference on Machine Learning, in Proceedings of Machine Learning Research 80:1861-1870 Available from https://proceedings.mlr.press/v80/haarnoja18b.html.

---

### Review · Reviewer_dn15 · 2025-01-25

**Summary Of Contributions:**

The paper introduces the Illustrated Landmark Graph (ILG), a novel approach for long-horizon policy learning. ILG bridges the gap between reinforcement learning (RL) and imitation learning by allowing human teachers to specify tasks as a graph of intermediate "landmarks," which are illustrated with examples. The proposed learning algorithm, ILG-Learn, leverages these graphs to iteratively plan and learn policies while incorporating teacher feedback through queries. This method reduces the need for manually specifying dense reward functions or providing full-length demonstrations. Experimental results demonstrate that the proposed approach outperforms standard Behavior Cloning and RCE, an example-based control algorithm.

**Audience:**

Yes

**Broader Impact Concerns:**

None.

**Claims And Evidence:**

Yes

**Requested Changes:**

Desired Adjustments:

- Provide baseline results for one more imitation learning method (ideally a recent one) and one more example-based or goal-conditioned RL approach.
- Provide more details on the human interaction. Was the "human" a scripted or synthetic agent, one of the authors, etc?
- Compare and contrast ILG-Learn with hierarchical reinforcement learning methods in the related work.
- Discuss more advanced versions of DAgger such as "ThriftyDAgger: Budget-Aware Novelty and Risk Gating for Interactive Imitation Learning" Hoque et al. CoRL 2021 in related work since DAgger is quite old and there have been many improvements since then.

Typos:
- page 3 "bahavior cloning"

**Strengths And Weaknesses:**

Strengths:
+ Innovative Task Specification: ILG provides a flexible and interpretable way to define long-horizon tasks, balancing abstraction and concreteness.
+ Empirical Evidence: Robust experiments on various tasks illustrate the efficacy of ILG-Learn over baselines like RCE and behavior cloning.
+ Reduced Human Effort: By focusing on intermediate landmarks, ILG reduces the need for detailed reward engineering or exhaustive demonstrations.
+ Scalability: The framework accommodates branching paths, allowing exploration of multiple strategies. This gives ILG the ability to incorporate multi-modality by having different paths in the graph that achieve the same goal.
+ General framework that seems to subsume goal-conditioned RL as a special case where there is just one sink node in the graph. I would recommend highlighting this fact more if this statement is correct.

Weaknesses:
- Teacher Burden: While reduced compared to traditional methods, selecting appropriate landmarks and densities may still require expertise.
- Limited Generalization: The method's performance relies heavily on task-specific ILG configurations, potentially limiting generalizability. However, I like the authors ideas for using foundation models to address this in the future.
- Baselines: Only two baselines is a bit concerning given that there are many types of BC and goal-conditioned RL that could be applied.
- Human Factors: The paper does not run a user study to explore human factors. I think this is fine for this type of paper that introduces a new algorithmic approach, but having some notion of how usable this method is in practice would significantly strengthen the paper and could be mentioned more in future work.

---

> ### Author Response · Authors · 2025-02-06
> **Response to Reviewer dn15 (part 1)**
>
> We thank the reviewer for their detailed review.
>
> **Provide baseline results for one more imitation learning method (ideally a recent one) and one more example-based or goal-conditioned RL approach**
>
> We thank the reviewer for the suggestion and have included Diffusion Policy as a more recent imitation learning baseline. We find that DiffusionPolicy’s performance is comparable to that of our original BC baseline, with moderately better performance on the StackAB task (0.28 success rate). While this supports our claim that ILG-Learn is a promising alternative imitation learning, we would like to note that comparison with state of the art policy architecture and optimization is not our primary focus. Rather, our experiments demonstrate that our specification and training pipeline allows a human teacher to facilitate long-horizon policy learning without reward engineering, demonstration, or the need to predict which approaches are most feasible for the learner. We envision future work that augments ILG-Learn with more modern policy architectures and optimization to enhance the robustness and sample efficiency of ILG-Learn, and identify this promising direction in the discussion section (Section 6).
>
> We also added Hierarchical Reinforcement Learning for Reward Machines (HRM) [8] as an additional baseline. HRM is an instantiation of the options framework for hierarchical RL that interleaves training the high-level policy over options and the low-level intra-option policies. Moreover, the learner receives structured rewards from a human-specified reward machine that exposes the high-level structure of the task to the learner. In this respect, HRM incorporates aspects of GCRL: each intra-option policy receives an additional reward when it triggers a transition to its desired reward machine state. This structure is similar to the ILG; HRM is unique among our baselines because it is well-suited to tasks such as StackChoice-Outwardview that require high-level planning. We manually design reward machines that are inspired by the reward machines used in the original HRM paper and the original reward of the Robosuite “Stack” benchmark [9]. This enabled HRM to achieve a high success rate in DiagonalMaze3x3.  Although we are still running some HRM baseline experiments due to initial technical difficulties, preliminary results suggest that HRM can complete the initial “grasp” part of both the StackAB task and StackChoice-OutwardView, but may not be able to fully complete the Stack. If our final evaluation confirms this negative result, that will help show that exposing the structure of a task using a reward machine does not alleviate the need for careful reward engineering and lends evidence that ILG-Learn may be a promising alternative to manual reward specification for long horizon tasks.
>
> **Provide more details on the human interaction. Was the "human" a scripted or synthetic agent, one of the authors, etc**
>
> The “human” interaction was scripted and is assumed to provide accurate and optimal feedback. We have added a precise description of our scripted human’s notion of ``success’’ for each landmark in our experiments in Appendix E: Environment Details. We also include Table 3 in the appendix, which details the hyperparameters that determine the frequency of scripted human interaction. We added a reference to this appendix when we introduce ILG-Learn’s “Success Estimation” subroutine in Section 4: Learning Algorithm.
>
> **Teacher Burden: While reduced compared to traditional methods, selecting appropriate landmarks and densities may still require expertise. … Human Factors: The paper does not run a user study to explore human factors.**
>
> We agree that designing an ILG and selecting ILG-Learn’s hyperparameters may require expertise and have updated the discussion section (Section 6) to identify a detailed user study (in the style of [3-5]) as an important direction for future work. In the context of short-horizon tasks, prior work on example based control suggests that providing goal  examples (analogous to our illustrative observations) is often much easier than providing full-length demonstrations or engineering a reward function [6,7], although the relative difficulty of each approach varies depending on the robot morphology and desired task. We believe that ILG-Learn allows a human to combine example-based control with a ``common sense’’ task decomposition, and would like to study the extent to which ILG design truly is intuitive and effective for a suite of long horizon tasks.

---

> > ### Author Response · Authors · 2025-02-06
> > **Response to Reviewer dn15 (part 2)**
> >
> > **Compare and contrast ILG-Learn with hierarchical reinforcement learning methods in the related work. … Discuss more advanced versions of DAgger such as "ThriftyDAgger”**
> >
> > We thank the reviewer for this suggestion. We have now expanded our related work section to include more discussion on closely related Hierarchical RL and Task and Motion Planning literature. Additionally, we have expanded on the human-in-the-loop policy learning section of the related work to include more discussions on relevant recent literature such as “ThriftyDAgger” [10].
> >
> > **Limited Generalization: The method's performance relies heavily on task-specific ILG configurations, potentially limiting generalizability.**
> >
> > We acknowledge that ILG-Learn requires the teacher to define an ILG for each long horizon task, and we believe that this shared abstraction is necessary for the teacher to provide structured guidance to the learner. We also believe that the ILG provides an interface through which it becomes easy to involve a foundation model: both for ILG construction (based on a natural language task description [11]) and to play the role of the teacher during training. We are excited to pursue this direction in future work.
> >
> > **General framework that seems to subsume goal-conditioned RL as a special case where there is just one sink node in the graph.**
> >
> > We believe that our framework does not subsume goal-conditioned reinforcement learning (GCRL). The GCRL framework in full generality allows the human to define an arbitrary set of goals. For example, a prominent GCRL paradigm takes the set of MDP states to be the set of goals, and trains a goal-conditioned policy via hindsight relabeling over collected experiences [1,2]. In contrast, our framework requires the human to specify a finite set of landmarks that serve as intermediate goals. Our framework can be viewed as an instantiation of GCRL where the desired final goal is the union of the ILG’s final landmarks.
> >
> > ---
> >
> > ### References
> >
> > [1] Marcin Andrychowicz, Filip Wolski, Alex Ray, Jonas Schneider, Rachel Fong, Peter Welinder, Bob McGrew, Josh Tobin, Pieter Abbeel, and Wojciech Zaremba. 2017. Hindsight experience replay. In Proceedings of the 31st International Conference on Neural Information Processing Systems (NIPS'17). Curran Associates Inc., Red Hook, NY, USA, 5055–5065.
> >
> > [2] Ofir Nachum, Shixiang Gu, Honglak Lee, and Sergey Levine. 2018. Data-efficient hierarchical reinforcement learning. In Proceedings of the 32nd International Conference on Neural Information Processing Systems (NIPS'18). Curran Associates Inc., Red Hook, NY, USA, 3307–3317.
> >
> > [3] Biyik, E., Palan, M., Landolfi, N.C., Losey, D.P. &amp; Sadigh, D.. (2020). Asking Easy Questions: A User-Friendly Approach to Active Reward Learning. Proceedings of the Conference on Robot Learning, in Proceedings of Machine Learning Research. 100:1177-1190 Available from https://proceedings.mlr.press/v100/b-iy-ik20a.html.
> >
> > [4]  Fitzgerald, T., Koppol, P., Callaghan, P., Wong, R.Q.J.H., Simmons, R., Kroemer, O. &amp; Admoni, H.. (2023). INQUIRE: INteractive Querying for User-aware Informative REasoning. Proceedings of The 6th Conference on Robot Learning, in Proceedings of Machine Learning Research 205:2241-2250 Available from https://proceedings.mlr.press/v205/fitzgerald23a.html.
> >
> > [5] Cui, Y., Koppol, P., Admoni, H., Simmons, R.G., Steinfeld, A., & Fitzgerald, T. (2021). Understanding the Relationship between Interactions and Outcomes in Human-in-the-Loop Machine Learning. International Joint Conference on Artificial Intelligence.
> >
> > [6] Benjamin Eysenbach and Sergey Levine and Ruslan Salakhutdinov. Replacing Rewards with Examples: Example-Based Policy Search via Recursive Classification. NeurIPS (2021).
> >
> > [7] Kevin Li and Abhishek Gupta and Vitchyr H. Pong and Ashwin Reddy and Aurick Zhou and Justin Yu and Sergey Levine. Reinforcement Learning with Bayesian Classifiers: Efficient Skill Learning from Outcome Examples. OpenReview. 2021.
> >
> > [8] Rodrigo Toro Icarte, Toryn Q. Klassen, Richard Valenzano, and Sheila A. McIlraith. 2022. Reward Machines: Exploiting Reward Function Structure in Reinforcement Learning. J. Artif. Int. Res. 73 (May 2022). https://doi.org/10.1613/jair.1.12440
> >
> > [9] Yuke Zhu and Josiah Wong and Ajay Mandlekar and Roberto Martin-Martin and Abhishek Joshi and Kevin Lin and Soroush Nasiriany and Yifeng Zhu. robosuite: A Modular Simulation Framework and Benchmark for Robot Learning. 2020.
> >
> > [10] Hoque, R., Balakrishna, A., Novoseller, E., Wilcox, A., Brown, D.S. &amp; Goldberg, K.. (2022). ThriftyDAgger: Budget-Aware Novelty and Risk Gating for Interactive Imitation Learning. Proceedings of the 5th Conference on Robot Learning, in Proceedings of Machine Learning Research 164:598-608 Available from https://proceedings.mlr.press/v164/hoque22a.html.
> >
> > [11] Jason Xinyu Liu, Ziyi Yang, Ifrah Idress, Sam Liang, and Benjamin Schornstein. Grounding Complex Natural Language Commands for Temporal Tasks in Unseen Environments. CoRL 23.

---

> > > ### Comment · Reviewer_dn15 · 2025-02-13
> > >
> > > Thank you for your responses and the updated experiments. I think the new experiments and updated discussion greatly improve the paper.

---

### Author Response · Authors · 2025-02-06
**global response**

We thank all the reviewers for their thoughtful and detailed reviews. We are excited that all the reviewers found our proposed method as a novel and innovative task specification for long-horizon tasks. In this global response, we would like to highlight additional experiments and updates to our manuscript based on all the feedback received.

### Experiments:
- **Additional Baselines.**
   - In response to Reviewers dn15, XYTR, and 3rxy, we added hierarchical reinforcement learning with reward machines (HRM) [1] as a baseline. HRM is an instantiation of the options framework for hierarchical RL [2] that interleaves training the high-level policy over options and the low-level intra-option policies. Moreover, the learner receives structured rewards from a human-specified reward machine that exposes the high-level structure of the task to the learner (More details on the setup are provided in Section 5 and Appendix D.4). Although we are still running some HRM baseline experiments, preliminary results suggest that HRM achieves lower success rates than ILG-Learn for long-horizon tasks, which strengthens ILG-Learn’s position as a promising alternative to manual reward specification for long-horizon tasks. We plan to provide an update here once the results for the HRM baseline are finalized.
   - In response to Reviewer dn15, we added Diffusion Policy [3] as an additional, more modern, behavior cloning baseline. We find that Diffusion Policy performs better than our original BC implementation, but achieves lower success rates than ILG-Learn on the block stacking (StackAB) and point maze navigation tasks, supporting our claim that ILG-Learn is a promising approach for long-horizon tasks.

- **Empirical investigation of data dependence.**
  - In response to reviewer XYTR, we conducted additional experiments in which we manipulated the quantity of human-provided data available to ILG-Learn, RCE, and behavior cloning. We added the results of this study to Table 1 in Section 5 (with more data presented in Table 4 in Appendix). We found that providing 1000 goal examples to RCE and 1000 demonstrations to BC yields moderately improved success rates (0.392 and 0.28, respectively) but neither baseline matches the 0.776 success rate achieved by ILG-Learn with just 10 illustrative observations per landmark. Suggesting that the hierarchical prior that ILG provides is an important ingredient in learning long-horizon tasks.

### Key Updates
-  In response to feedback from Reviewers 3rxy, we have revised our Abstract and Introduction (Section 1) to improve the clarity of our contributions: ILG specification and ILG-Learn algorithm.
-  In response to feedback from Reviewers 3rxy; dn15, we expand on the related work (Section 2) to compare and contrast the proposed method with Hierarchical RL, Task and Motion Planning, automata-based specification.
- In response to feedback from Reviewers dn15; XYTR, we have updated our discussion section (Section 6) to clearly state the limitations of our proposed approach and discuss concrete directions for future work highlighting our plans to incorporate foundation models to reduce the burden on the human teacher.
- We thank all the reviewers for other thoughtful feedback on the presentation. We have suitably updated the text, figures, tables, and algorithms throughout the paper.

### References
[1] Rodrigo Toro Icarte, Toryn Q. Klassen, Richard Valenzano, and Sheila A. McIlraith. 2022. Reward Machines: Exploiting Reward Function Structure in Reinforcement Learning. J. Artif. Int. Res. 73 (May 2022). https://doi.org/10.1613/jair.1.12440

[2] Richard S. Sutton, Doina Precup, Satinder Singh. Between MDPs and semi-MDPs: A framework for temporal abstraction in reinforcement learning. Artificial Intelligence, Volume 112, Issues 1–2, 1999.

[3] Cheng Chi, Zhenjia Xu, Siyuan Feng, Eric Cousineau, Yilun Du, Benjamin Burchfiel, Russ Tedrake, & Shuran Song (2024). Diffusion Policy: Visuomotor Policy Learning via Action Diffusion. The International Journal of Robotics Research

---

> ### Author Response · Authors · 2025-02-07
> **HRM Baseline Results**
>
> **Additional Baselines**
>
> To follow up on our earlier comment, the HRM [1] baseline experiments have completed and we have updated Table 1 accordingly. For our StackAB task, we defined a reward machine that provides dense rewards inspired by the original reward function of robosuite's [2] ``Stack'' benchmark. We found that HRM achieved a mean success rate of 0.005 over five random seeds, significantly lower than the 0.776 success rate achieved by ILG-Learn. For the DiagonalMaze tasks, we found that HRM achieved high success rate for the 3x3 variant, but not the larger mazes. This confirms our preliminary results, and continues to suggest that ILG-Learn may be a promising alternative to manual reward design for long-horizon tasks.
>
> ---
> ### References
> [1] Rodrigo Toro Icarte, Toryn Q. Klassen, Richard Valenzano, and Sheila A. McIlraith. 2022. Reward Machines: Exploiting Reward Function Structure in Reinforcement Learning. J. Artif. Int. Res. 73 (May 2022). https://doi.org/10.1613/jair.1.12440
>
> [2] Yuke Zhu and Josiah Wong and Ajay Mandlekar and Roberto Martin-Martin and Abhishek Joshi and Kevin Lin and Soroush Nasiriany and Yifeng Zhu. robosuite: A Modular Simulation Framework and Benchmark for Robot Learning. 2020.

---

> ### Author Response · Authors · 2025-02-13
> **Second revision**
>
> We thank the anonymous reviewers for their responses and have posted an updated manuscript with the following changes, in order of priority:
> - We updated the learning curves in Appendix F to use color to show individual random seeds, and used dashed lines to show the early termination behavior of ILG-Learn.
> - While updating the learning curves, we noticed that some ILG-Learn experiments were originally run with an option that disabled the termination feature of ILG-Learn that is described in Section 4 and Appendix A. We re-ran these experiments and updated Table 1 and Table 4. The success rate for ILG-Learn on DiagonalMaze5x5 increased from 0.991 to 1.00, and for StackAB increased from 0.776 to 0.975.  We thank the reviewers for bringing this to our attention.
> - We noticed that we reported the mean success rate of ILG-Learn (illustrationCount=10) for DiagonalMaze7x7 over 10 random seeds, rather than 5 as we did for all other experiments. We updated Table 1 and Table 2 to only reflect the first (chronologically by experiment timestamp) 5 random seeds. This changed the mean success rate inconsequentially (from 0.971 to 0.995).
> - Minor: We updated Section 4 to define ILG-Learn’s IntervalLength hyperparameter in terms of environment steps rather than episodes. Since the episode horizon is fixed these notions are equivalent, however elsewhere (e.g. in Table 3 in Appendix B) we report intervalLength in terms of environment steps. We also added a paragraph in Section 4 that directs the reader to Appendix B for discussion of hyperparameters.
> - Minor: We fixed a mis-reported hyperparameter value in Appendix D.1: we used the “max” as the Q-combinator for the RCE subroutine in all our experiments
> - Minor: We added a sentence in Section 2 comparing our approach with [1], we improved the wording of the description of RCE in Section 4, and we improved the wording of the latter paragraphs of Section 6.
> -----
>
> ## Reference:
> [1] Beyazit Yalcinkaya, Niklas Lauffer, Marcell Vazquez-Chanlatte, and Sanjit A. Seshia, “Compositional Automata Embeddings for Goal-Conditioned Reinforcement Learning,” NeurIPS, 2024.

---

### Decision · Action_Editor_JFux · 2025-02-22

**Recommendation:** Accept as is

**Comment:**

This paper studies the use of landmark graphs as a lighter approach for humans to provide demonstrations in sequential decision-making problems (landmarks instead of complete trajectories). The benefits of such an approach go beyond the fact that it makes it less burdensome for humans to provide demonstrations; it also, for example, allows for a modularization of the policy learning problem, as policies are learned for specific edges of the graph.

As can be seen in the initial reviews, the reviewers had concerns about the presentation of the paper (clarity, lack of details on some aspects, a more careful discussion about related work) and the lack of representative baselines. The discussion with one of the reviewers even revealed a bug in one particular set of results! The authors addressed these issues during the rebuttal, and the reviewers all agreed that the new version is substantially improved and should be accepted.

**Audience:**

All reviewers agreed from the beginning that this was an interesting topic of research and that the study was interesting.

**Claims And Evidence:**

Yes, initially, the reviewers considered the claims too broad and the evidence insufficient to support these claims, primarily due to the lack of some baselines and further discussion about related work. The authors addressed these concerns in the rebuttal phase.